# Silencing of SRRM4 suppresses microexon inclusion and promotes tumor growth across cancers

Sarah A. Head [1]*, Xavier Hernandez-Alias [1], Jae-Seong Yang [1,2], Ludovica Ciampi [1], Violeta Beltran-Sastre [1], Antonio Torres-Méndez [1], Manuel Irimia [1,3,4], Martin H. Schaefer [1,5]*, Luis Serrano [1,3,4]*

1 Centre for Genomic Regulation (CRG), The Barcelona Institute of Science and Technology, Barcelona, Spain, 2 Centre de Recerca en Agrigenòmica, Consortium CSIC-IRTA-UAB-UB, Cerdanyola del Vallès, Barcelona, Spain, 3 Universitat Pompeu Fabra (UPF), Barcelona, Spain, 4 ICREA, Barcelona, Spain, 5 IEO European Institute of Oncology IRCCS, Department of Experimental Oncology, Milan, Italy

* sarah.dibartolo@crg.eu (SAH); martin.schaefer@ieo.it (MHS); luis.serrano@crg.eu (LS)

**Data Availability Statement:** Raw RNA sequencing data generated in this study have been deposited in the NCBI Sequence Read Archive (SRA) with accession numbers PRJNA474911 and

## Abstract

RNA splicing is widely dysregulated in cancer, frequently due to altered expression or activity of splicing factors (SFs). Microexons are extremely small exons (3–27 nucleotides long) that are highly evolutionarily conserved and play critical roles in promoting neuronal differentiation and development. Inclusion of microexons in mRNA transcripts is mediated by the SF Serine/Arginine Repetitive Matrix 4 (SRRM4), whose expression is largely restricted to neural tissues. However, microexons have been largely overlooked in prior analyses of splicing in cancer, as their small size necessitates specialized computational approaches for their detection. Here, we demonstrate that despite having low expression in normal non-neural tissues, SRRM4 is further silenced in tumors, resulting in the suppression of normal microexon inclusion. Remarkably, SRRM4 is the most consistently silenced SF across all tumor types analyzed, implying a general advantage of microexon down-regulation in cancer independent of its tissue of origin. We show that this silencing is favorable for tumor growth, as decreased SRRM4 expression in tumors is correlated with an increase in mitotic gene expression, and up-regulation of SRRM4 in cancer cell lines dose-dependently inhibits proliferation in vitro and in a mouse xenograft model. Further, this proliferation inhibition is accompanied by induction of neural-like expression and splicing patterns in cancer cells, suggesting that SRRM4 expression shifts the cell state away from proliferation and toward differentiation. We therefore conclude that SRRM4 acts as a proliferation brake, and tumors gain a selective advantage by cutting off this brake.

## Introduction

Alternative splicing (AS) is an important mechanism for increasing the complexity of the human genome, allowing one gene to perform different specialized functions in different cellular or developmental contexts. The most common type of AS in mammalian pre-mRNA is

PRJNA551123. The complete tables of gene expression and exon inclusion quantification from these datasets, as well as raw exon inclusion tables from the TCGA datasets analyzed in this study, have been deposited in the Synapse data repository (http://www.synapse.org; Synapse ID: syn21965151; doi: 10.7303/syn21965151). All other data underlying the figures in this manuscript are included in supplementary files as indicated in the figure legends.

**Funding:** This project was funded in part by a grant from the Plan Estatal de Investigación Científica y Técnica y de Innovación to L.S. (PGC2018-101271-B-I00, http://www.ciencia.gob.es). S.A.H. is supported by a Marie Skłodowska-Curie Individual Fellowship from the European Union's Horizon 2020 research and innovation programme (MSCA-IF-2017-794629, http://ec.europa.eu/). X.H. is supported by a PhD fellowship from the Fundación Ramón Areces (http://www.fundacionareces.es). We acknowledge the support of the Spanish Ministry of Economy and Competitiveness, 'Centro de Excelencia Severo Ochoa', the CERCA Programme / Generalitat de Catalunya, and the Spanish Ministry of Economy, Industry and Competitiveness (MEIC) to the EMBL partnership. The funders had no role in study design, data collection and analysis, decision to publish, or preparation of the manuscript.

**Competing interests:** The authors have declared that no competing interests exist.

**Abbreviations:** AS, alternative splicing; BDNF, brain-derived neurotrophic factor; Ct, cycle threshold; DM, deletion mutant; eMIC, enhancer of microexons; ESC, embryonic stem cell; EV, empty vector; FDR, false discovery rate; GO, gene ontology; IVC, individually ventilated cage; MI, mitotic index; NE, neuroendocrine; PSI, percent spliced in; PT, primary tumor; qPCR, quantitative PCR; RA, retinoic acid; REST, RE1-silencing transcription factor; RNA-seq, RNA sequencing; RSEM, RNA-Seq by Expectation-Maximization; RT, room temperature; RT-PCR, reverse transcription PCR; SF, splicing factor; SRRM4, Serine/Arginine Repetitive Matrix 4; STN, solid tissue normal; TCGA, The Cancer Genome Atlas; TNM, tumor/ node/metastasis; WT, WILD-TYPE.

exon skipping, in which a cassette exon is either retained or removed from the mature mRNA transcript. The average exon in humans is approximately 140 nucleotides long and contains features that are recognized by splicing factors (SFs), which bind both inside and outside the exonic sequence to catalyze the splicing reaction.

Microexons, defined herein as exons between 3 and 27 nucleotides in length, have recently been shown to comprise a distinct functional class of cassette exons with higher evolutionary conservation, open reading frame preservation, and enriched localization within protein interaction domains compared with their longer counterparts [1,2]. In contrast to normal exons, microexons are generally too short to contain standard exonic splicing enhancers and thus require a specialized machinery to facilitate their recognition and inclusion. This is mediated primarily by the SF Serine/Arginine Repetitive Matrix 4 (SRRM4, aka nSR100), which has been reported to be expressed at high levels only in neurons [1,3]. SRRM4 plays a critical role in regulating neuronal differentiation, as knockdown and knockout experiments have revealed morphological and functional deficits in cultured neurons as well as the nervous systems of zebra danio and mice [3–7]. In addition, SRRM4 was shown to be down-regulated in the brains of some autistic patients, resulting in a global misregulation of microexon splicing [1], further implicating SRRM4 as a critical factor in brain development. However, the role of microexons in nonneural tissues, if any, has been relatively unexplored to date.

Dysregulation of AS has been implicated in numerous human diseases, including cancer [8]. Loss of splicing fidelity is extremely common in cancer, due either to mutations that directly affect splice sites or regulatory regions within pre-mRNAs or alterations in SF expression or activity [9–11]. These changes may result in the expression of protein isoforms that confer selective advantages to cancer cells, either by increasing tumorigenic activity or decreasing tumor suppressive activity. Large-scale studies of AS alterations in cancer have been aided greatly in recent years by consortia such as The Cancer Genome Atlas (TCGA), which has collected multiomic data from thousands of patient samples. The availability of raw RNA sequencing (RNA-seq) data from both tumor and corresponding normal tissues has facilitated comparative analyses of SF expression and AS, revealing widespread dysregulation of splicing in cancer [12–16]. However, microexons are usually systematically ignored in such analyses, due to their small size and the resulting difficulty in detecting and separating them from background noise. Special computational approaches are therefore required to accurately quantify microexon inclusion [1,17–19].

Here, we use public data from TCGA to analyze changes in SRRM4 expression and microexon inclusion between tumor and normal samples from 9 different tissues (Fig 1). Surprisingly, we find that not only are SRRM4 and its microexon program globally down-regulated in cancer despite having low basal inclusion in normal nonneural tissues, but also in fact SRRM4 is the most consistently down-regulated across tumor types out of all SFs analyzed. We map this decreased expression to a strong increase in methylation of SRRM4 promoters, indicative of epigenetic silencing of SRRM4 expression in cancer. We show that SRRM4 expression and exon inclusion changes anticorrelate with mitotic gene expression in tumors, signifying an inverse relationship with cancer cell proliferation. Correspondingly, we observe marked inhibition of cancer cell proliferation with SRRM4 overexpression both in vitro and in a mouse xenograft model, accompanied by induction of neuron-like splicing and expression patterns. We conclude that the splicing program controlled by SRRM4 acts as a brake on proliferation by promoting differentiation and that tumors gain a proliferative advantage by cutting off this brake.

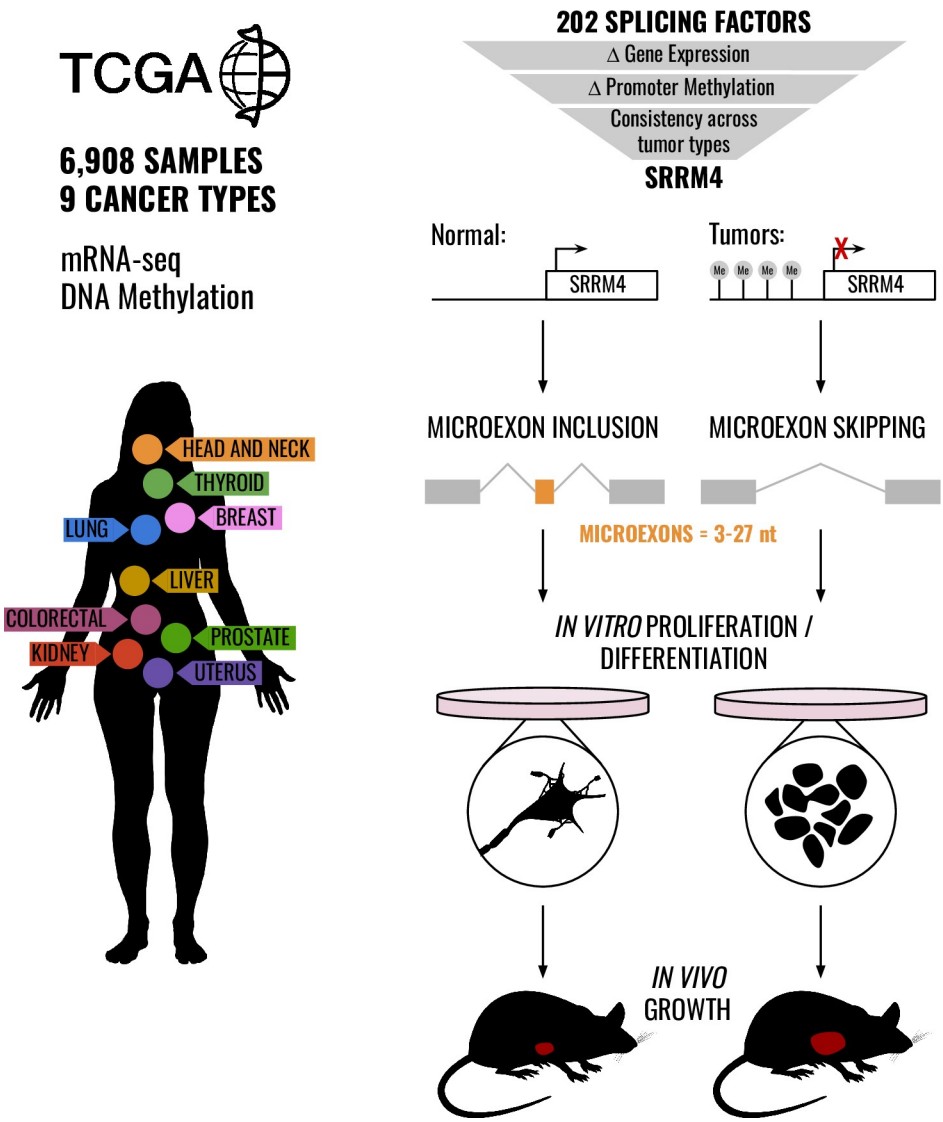

**Fig 1. Overview of the study.** We analyze SF dysregulation in tumors across 9 different tissues from TCGA. Out of 202 SFs, only SRRM4 is consistently silenced by promoter hypermethylation across cancers, resulting in suppressed microexon inclusion in tumors. SRRM4 expression in cancer cells leads to differentiated neuron-like splicing and expression patterns, accompanied by a decrease in cell proliferation in vitro and in a mouse xenograft model. mRNA-seq, mRNA sequencing; nt, nucleotide; SF, splicing factor; SRRM4, Serine/Arginine Repetitive Matrix 4; TCGA, The Cancer Genome Atlas.

## Results

### SRRM4 expression is silenced in cancer with high consistency across tumor types

Using publicly available, preprocessed RNA-seq data from TCGA, we first analyzed changes in SRRM4 expression between primary tumor (PT) and solid tissue normal (STN) samples. Nine tissue types were selected that had a sufficiently large number of tumor and matched normal samples to perform conclusive statistics ($N \geq 20$ each), as well as available raw RNA-seq files for quantification of splice variants (S1 Table). In accordance with the known neural specificity of SRRM4, we found that expression levels in the normal samples were generally low, with

median values ranging from 0.0 (liver) to 8.3 (thyroid) RNA-Seq by Expectation-Maximization (RSEM) normalized expression units. Remarkably, the median expression levels of SRRM4 were significantly lower in tumors from all tissues, with the exception of liver that had median expression of 0.0 in both tumor and normal samples ($p < 0.001$; Wilcoxon–Mann–Whitney test; Fig 2A, S1 Data; details of the statistical tests are found in the Methods section under "Statistical information"). To investigate the mechanism underlying this down-regulation, we used DNA promoter methylation array data to assess gene regulation at the epigenetic level [20], as promoter methylation is a well-established mechanism of gene silencing. Consistent with the down-regulated expression, we found an increase in SRRM4 promoter methylation that was highly significant in tumors from all tissues, even in liver ($p < 0.001$; Wilcoxon–Mann–Whitney test; Fig 2B, S2 Data). Taken together, these results suggest a global silencing of SRRM4 in cancers, beyond their normally low inclusion levels in nonneural tissues.

Despite the high frequency of AS alterations in cancer, the degree of consistency of SRRM4 silencing across tissues was particularly striking. Other members of the SRRM family (SRRM1–3) did not show the same pattern of consistent change (significant and in the same direction) across tissues at either the expression or the promoter methylation level (Fig 2C, S1 and S2 Data). In fact, from a list of 202 known SFs (see "Definition of splicing factor list" in Methods), only SRRM4 had significantly decreased expression and increased promoter methylation across all tumor types (Fig 2D and 2F, S1 and S2 Data). While 4 other SFs had consistently increased expression (SNRPB, RNPS1, RBM34, and ELAVL1), the absolute magnitude of the change in expression was higher for SRRM4 (mean fold change of 3.67×) compared with the other 4 (1.25 to 1.78×) (Fig 2D and 2E). At the promoter methylation level, one other SF was found to be hypermethylated in all tissues, and to a similar extent as SRRM4 (RALYL; Fig 2F and 2G), but this hypermethylation did not translate to decreased expression in all tissues. Taken together, these findings demonstrate a global and significant pattern of silencing of SRRM4 in cancer.

## Inclusion of SRRM4 target exons is decreased in tumors

To evaluate the effects of SRRM4 dysregulation at the level of microexon inclusion, we implemented a computational pipeline for the quantification and statistical evaluation of exon inclusion levels from RNA-seq data. Our pipeline makes use of *vast-tools* [21], a toolset for profiling AS from sequencing data that is capable of accurately quantifying inclusion levels (defined by "percent spliced in" or PSI) of exons as small as 3 nucleotides. We ran this pipeline on 6,264 PT and 644 STN samples from TCGA (S1 Table). Inclusion levels were quantified for 219,018 cassette exons, and, for each event, statistical comparisons were performed between tumor and corresponding normal samples in each of the 9 tissues. These results have been uploaded to VastDB [21] as a public resource (http://vastdb.crg.eu).

For each tissue type, we next defined a list of differentially spliced cassette exons between tumor and normal samples ($|\Delta PSI| > 5$ and q $< 0.01$, Wilcoxon–Mann–Whitney test; S3 Data). The resulting list comprised between 468 and 1,884 exons depending on the tissue type, of which between 6.6% (head and neck cancer) and 12.7% (colorectal cancer) were microexons (summary in S3 Data). These differentially spliced exons were significantly enriched for experimentally determined SRRM4 targets (S4 Data; Methods) in every cancer type tested ($p < 10^{-6}$; Fisher test).

As SRRM4 promotes inclusion of its target exons, we hypothesized that down-regulation of SRRM4 in tumors would lead to decreased inclusion levels of known SRRM4-regulated exons [22]. In support of this hypothesis, we found that over 70% of significantly changing SRRM4 target exons across tissues were decreased in tumors compared with normal samples (Fig 3A,

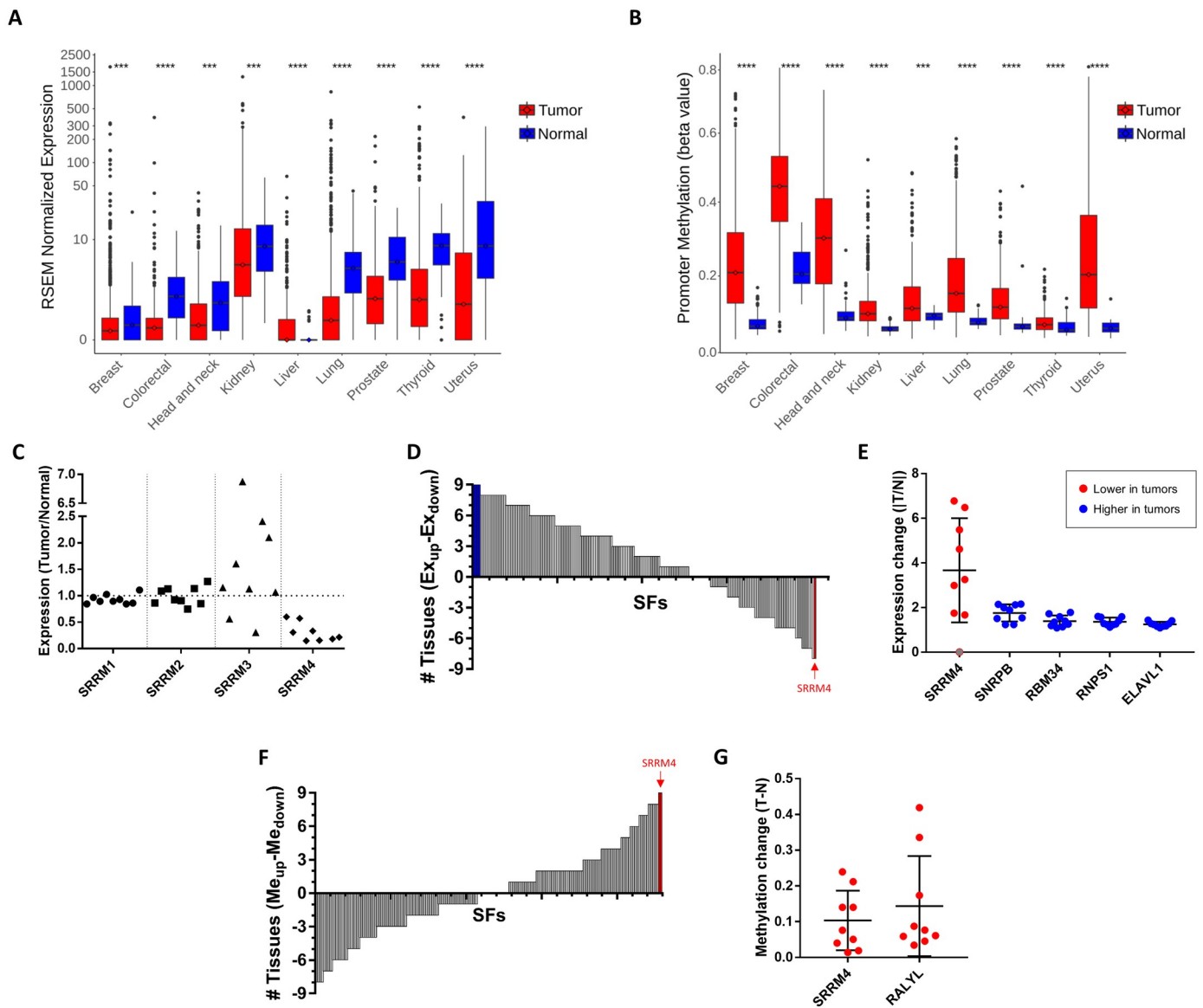

**Fig 2. Analysis of TCGA data reveals SRRM4 silencing in cancer across tissue types.** (A) Changes in SRRM4 expression between normal and tumor samples (2-tailed unpaired Wilcoxon–Mann–Whitney test; * $p < 0.05$, ** $p < 0.01$, *** $p < 0.001$, **** $p < 0.0001$. Boxes expand from the first to the third quartile, with the center values indicating the median. The whiskers define a confidence interval of median ± $1.58^*\text{IQR}/\sqrt{(n)}$). (B) Changes in SRRM4 promoter methylation between normal and tumor samples (2-tailed unpaired Wilcoxon–Mann–Whitney test; * $p < 0.05$, ** $p < 0.01$, *** $p < 0.001$, **** $p < 0.0001$. Boxes expand from the first to the third quartile, with the center values indicating the median. The whiskers define a confidence interval of median ± $1.58^*\text{IQR}/\sqrt{(n)}$). (C) SRRM4 is the only SRRM gene family member with consistently changing expression across tissues. Each point in the plot represents the ratio of median expression values between tumor and normal samples for 1 tissue. (D) The sum of the number of tissues with significantly up-regulated (positive) and down-regulated (negative) expression in tumors, for each of the 202 SFs. (E) Fold change in median expression of the 4 SFs that consistently increase across all tumor types (blue) and SRRM4, which decreases in tumors (red), with each point representing 1 tissue. SRRM4 expression in liver, which has a median value of 0.0, is shown in gray. Error bars represent SD. (F) The sum of the number of tissues with significantly up-regulated (positive) and down-regulated (negative) methylation in tumors, for each of the 202 SFs. (G) Fold change in median methylation (beta values) of the 2 SFs that consistently increase across all tumor types, with each point representing 1 tissue. Error bars represent SD. Data underlying Fig 2A and 2B can be found in S1 Data figures. Data underlying Fig 2C–2E can be found in S1 Data. Data underlying Fig 2F and 2G can be found in S2 Data. SF, splicing factor; SRRM4, Serine/Arginine Repetitive Matrix 4; TCGA, The Cancer Genome Atlas.

S1 and S2 Figs). This result is particularly remarkable given that around two-thirds of known SRRM4-regulated exons are not included in healthy nonneural tissues, meaning they cannot possibly decrease further in tumors (Fig 3B). Of the remaining one-third of SRRM4 target

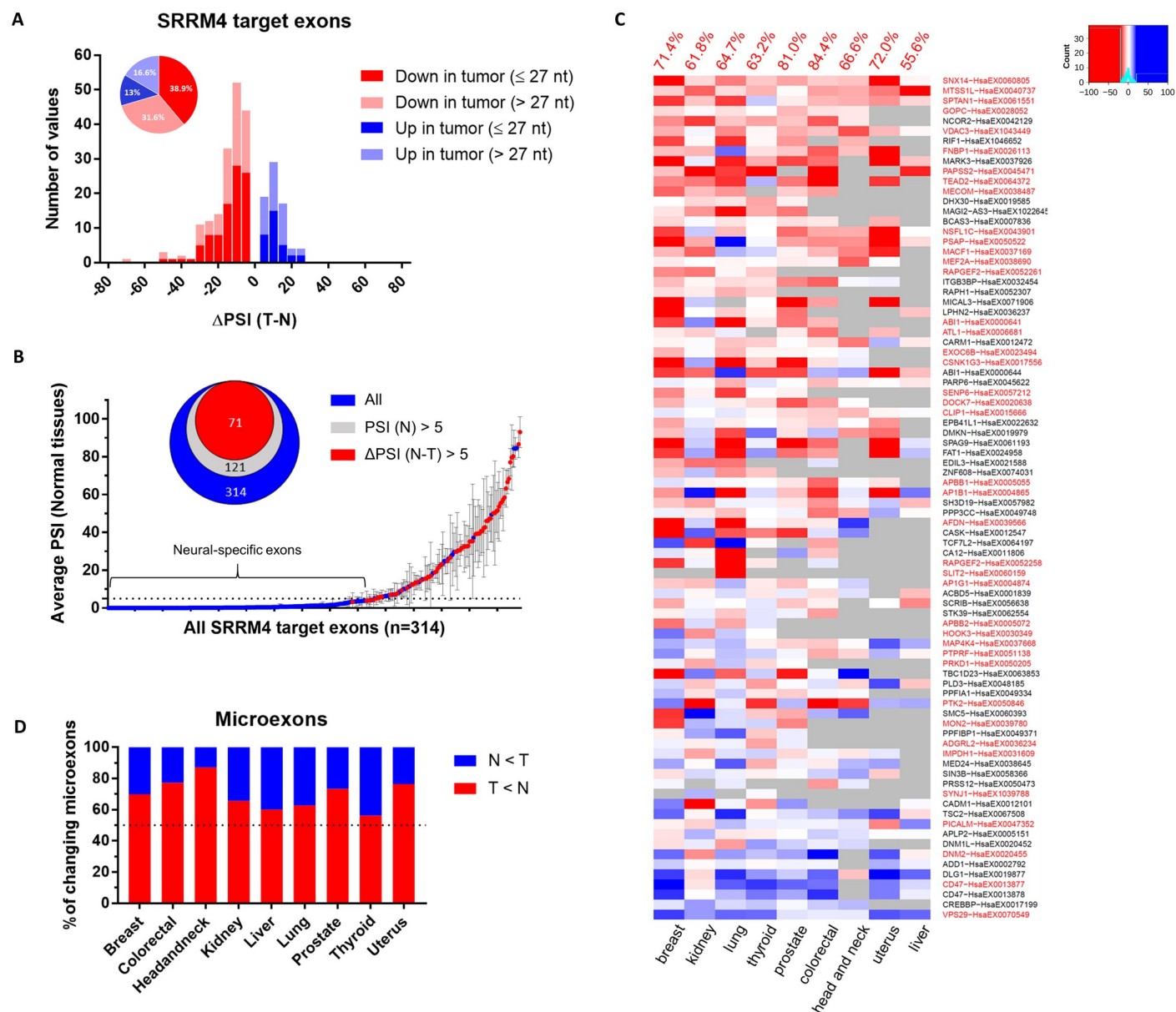

**Fig 3. Inclusion of SRRM4-regulated exons is decreased in tumors and cancer cell lines.** (A) Distribution of ΔPSIs of differentially spliced SRRM4 target exons between normal and tumor samples in the TCGA datasets. Only exons with significant changes (q < 0.01, |ΔPSI| ≥ 5) are shown. Microexons (≤27 nt) and non-microexons (>27 nt) are represented in different color shades as shown in the legend. (B) SRRM4 target exons with respect to their average PSI in normal tissues. Red points are exons with PSI decreasing by at least 5% in at least 1 tumor type. Error bars represent SD of all tissues with quantifiable inclusion levels for each exon. (C) Heatmap of SRRM4 target exons ΔPSIs (Tumor − Normal) across tissue types (red = lower in tumor, blue = higher in tumor). Microexons (≤27 nt) are highlighted in red along the y-axis. The percentage of SRRM4 target exons decreasing in each tumor type is indicated in red on the top. Gray boxes indicate exons without sufficient read coverage for quantification in at least 20 tumor and normal samples. (D) The proportion of differentially regulated microexons (≤27 nt) decreasing (red) vs. increasing (blue) in tumors from each tissue type. Only microexons with significant changes (q < 0.01, |ΔPSI| ≥ 5) are shown. Data underlying Fig 3A can be found in S3 and S4 Data. Data underlying Fig 3B and 3C can be found in S1 Data figures. Data underlying Fig 3D can be found in S3 Data. nt, nucleotide; PSI, percent spliced in; SRRM4, Serine/Arginine Repetitive Matrix 4; TCGA, The Cancer Genome Atlas.

exons with an average PSI > 5 in normal nonneural tissues, a majority (64%) were found to significantly decrease in at least 1 tumor type (Fig 3B, red points). For each individual tissue, the number of SRRM4-regulated exons significantly decreasing in tumors was also larger than

those increasing in tumors (Fig 3C). Using all significantly changing exons as a reference, the fraction of decreased SRRM4 target exons was larger than expected by chance in 5 of the 9 tissue types (lung, breast, prostate, uterus, and colorectal; $p < 0.05$; Fisher test). In agreement with the known role of SRRM4 as a key regulator of microexon splicing, we found that a majority of the differentially included SRRM4 targets were $\leq 27$ nucleotides in length, and the proportion significantly decreasing in tumors was larger than those increasing for every tumor type (Fig 3D). Furthermore, the SRRM4 targets with decreased average inclusion levels in tumors were more highly enriched in microexons (63%) as compared with those increasing in tumors (39%) (S3 Fig).

To further demonstrate the regulatory impact of SRRM4 activity on SRRM4 target inclusion levels, we correlated the expression levels of SRRM4 with the PSI of its target exons across TCGA tumor samples. The median Spearman correlation was larger than 0 in all cancer types (S4A Fig). To determine if the consistently positive correlation levels were larger than expected by chance, we generated background distributions of median correlations between the PSI of SRRM4 target exons and the expression levels of random genes. In 5 of the 9 cancer types, we found that the median correlation was larger than expected by chance (lung, breast, prostate, head and neck, and colorectal; $p < 0.05$; randomization test). Similar results, but in the opposite direction, were obtained when considering the methylation status of SRRM4 instead of its expression (S4B Fig). These observations demonstrate that the inclusion of SRRM4 target exons is generally down-regulated in tumors as a functional consequence of decreased SRRM4 expression.

Taken together, these results suggest that the observed down-regulation of microexon inclusion is a general phenomenon across a wide variety of cancer types. Indeed, a recent study of splicing changes between glia and glioblastoma also reported decreased microexon inclusion in glioblastoma [23], which is consistent with our findings in tumors from nonneural tissues and further suggests the generalizability of these findings to different cancer types.

## SRRM4 expression is inversely related to cell proliferation

To investigate the potential functional consequence of SRRM4 target exon down-regulation, we performed gene ontology (GO) term enrichment analysis [24] on the genes with decreased exon inclusion in tumors. The most significantly overrepresented biological processes among these genes were plasma membrane bounded cell projection organization, generation of neurons, cell morphogenesis involved in neuron differentiation, and neuron development (Fig 4A, S5 Data). These functions are consistent with the known role of SRRM4 in promoting neuronal differentiation and suggest that decreased expression of SRRM4 might decrease differentiation-related activities in tumors.

Because neuronal differentiation is accompanied by cell cycle withdrawal and cessation of cell division [25], we hypothesized that tumors might benefit from down-regulating these neuronal differentiation–related processes to gain a proliferative advantage. To assess proliferation in tumor samples, we monitored the mitotic index (MI), an mRNA expression signature that has been shown to correlate with the mitotic activity of cancer cells and is therefore used as an expression-based marker for cell proliferation [26]. In support of our hypothesis, we observed a negative correlation ($R_{Spearman} = -0.36$, $p < 2.2e-16$, Fig 4B) between the MI and SRRM4 expression in tumor samples, while a positive correlation was observed at the promoter methylation level ($R_{Spearman} = 0.45$, $p < 2.2e-16$, Fig 4C). This result suggested that SRRM4 expression might negatively affect the proliferation of tumor cells. Furthermore, for each individual SRRM4 exon, we calculated a Spearman correlation coefficient between the PSI of the exon and MI across tumor samples and found a strong negative association between the direction of

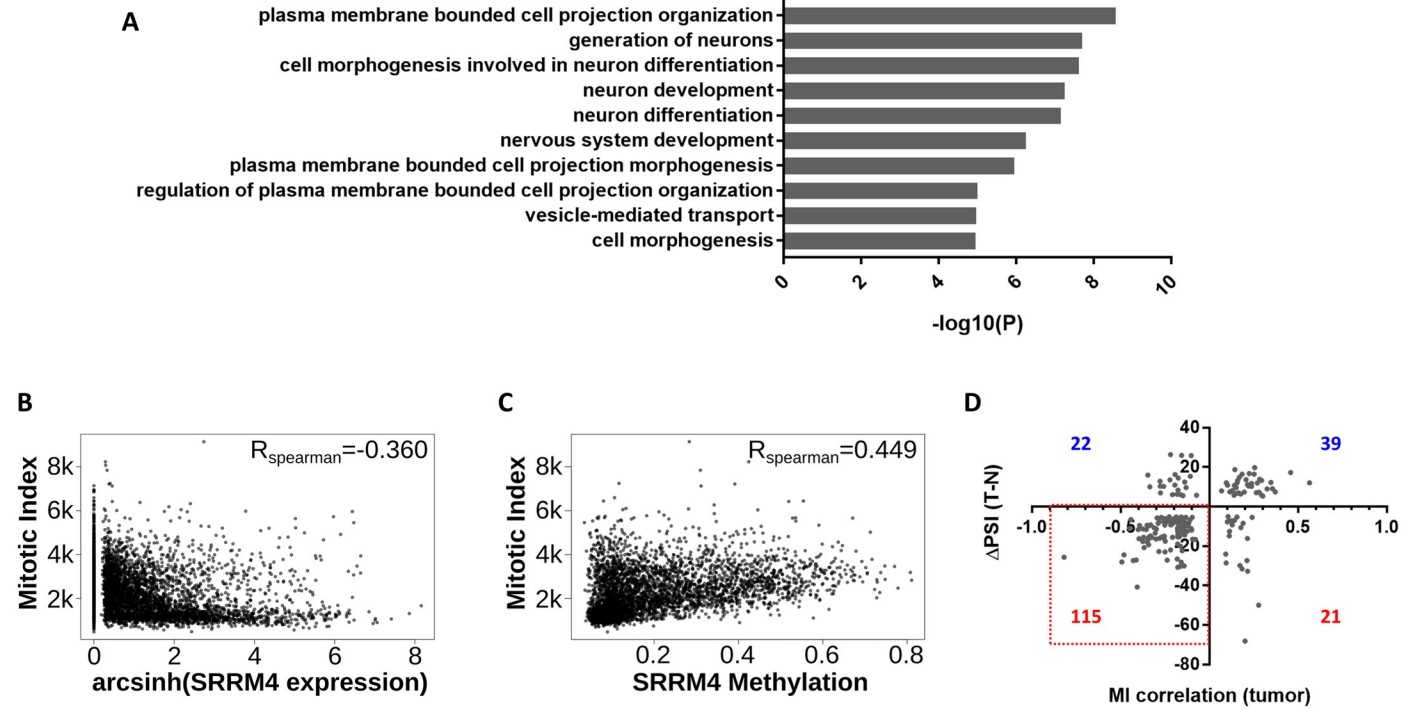

**Fig 4. SRRM4 correlates with MI in tumors.** (A) GO term enrichment of genes with down-regulated SRRM4 target exons in tumors from TCGA. The top 10 most significantly enriched terms are shown (GO biological process complete; 2-sided hypergeometric test with Bonferroni step down correction). (B) Spearman correlation of MI gene signature with SRRM4 expression across TCGA tumor samples. Each point represents 1 tumor sample. (C) Spearman correlation of MI gene signature with SRRM4 promoter methylation (beta value) across TCGA tumor samples. Each point represents 1 tumor sample. (D) x-Axis: Spearman correlation coefficient ($R_{Spearman}$) between MI gene signature and exon PSI for each SRRM4 target exon in tumors; y-axis: ΔPSI (Tumor − Normal). The number of exons in each quadrant is indicated (red indicates negative ΔPSI, and blue indicates positive ΔPSI). Each point represents 1 exon from 1 tissue. Only exons with |ΔPSI| > 5 and q < 0.01 are shown. Data underlying Fig 4A can be found in S5 Data. Data underlying Fig 4B–4D can be found in S1 Data figures. GO, gene ontology; MI, mitotic index; PSI, percent spliced in; SRRM4, Serine/Arginine Repetitive Matrix 4; TCGA, The Cancer Genome Atlas.

this correlation for each exon and its ΔPSI in tumors (*p* = 3.68e-16, binomial test). In agreement, the vast majority (83.9%) of SRRM4 target exons with a negative MI correlation in tumors from a given tissue also had significantly decreased inclusion in tumors from that tissue (Fig 4D), supporting the hypothesis that decreased inclusion of these exons is likely to be associated with increased proliferative activity. Furthermore, a majority of the SRRM4 target exons with significant negative MI correlations were ≤27 nucleotides in length, while those with positive correlations tended to be longer (S5 Fig). This finding suggests that SRRM4 target microexons may contribute more to the antiproliferative phenotype than longer SRRM4 target exons; however, we cannot rule out the possibility that longer SRRM4 targets also contribute.

We further tested whether SRRM4 expression correlates with other tumor features apart from proliferation. We did not observe any clear trend of SRRM4 expression changes across different tumor/node/metastasis (TNM) stages (S6A Fig) nor was there a significant correspondence of SRRM4 expression levels with patient survival in any tissue type (Kaplan–Meier method, q < 0.05, false discovery rate [FDR] corrected). There was also no correlation with tumor purity, as measured by 2 different methods: ESTIMATE [27] ($R_{Spearman}$ = −0.04) and CPE [28] ($R_{Spearman}$ = 0.038) (S6B Fig). We did detect significant associations between SRRM4 expression and mutations in certain cancer genes, where the presence of mutations in those genes are generally associated with a lower expression of SRRM4. In particular, out of 15

cancer genes tested, 10 of them raised significant associations ($p < 0.05$), with the presence of 8 of them being associated with a lower SRRM4 expression (S6C Fig). These observations are consistent with the association of most cancer gene mutations with MI, which in 9/10 cases change in the opposite direction as SRRM4 (S6D Fig).

## SRRM4 overexpression inhibits cancer cell growth in vitro and in vivo

To test the hypothesis that SRRM4 expression affects cancer cell proliferation, we generated several stable cell lines with tetracycline-inducible expression of SRRM4. Six different commonly used cancer cell lines were chosen: HeLa (cervical cancer), MCF7 (estrogen receptor–positive breast cancer), MDA-MB-231 (triple negative breast cancer), HCT116 (colon cancer), DU145 (prostate cancer), and SH-SY5Y (neuroblastoma). We also generated control cell lines using empty vector (EV) and an inactive SRRM4 deletion mutant (DM) missing 39 amino acids in the carboxyl terminus enhancer of microexons (eMIC) domain essential for splicing activity [22]. In all 6 cases, the addition of increasing concentrations of doxycycline to the cells dose-dependently inhibited cell growth only in the wild-type (WT) SRRM4 expressing cells, and not in the EV or DM control lines (Fig 5A–5D, S7 and S8 Figs). Similar antiproliferative effects were observed in the non-tumor-derived, immortalized kidney cell line, HEK293 (S7 and S8 Figs). These results demonstrate that increased SRRM4 expression leads to decreased cell proliferation across cell lines. These antiproliferative effects are mediated by SRRM4 SF activity, as evidenced by the fact that the inactive mutant has no such effects.

Although the antiproliferative effects of SRRM4 are evident in this 4-d growth assay, we observed that the levels of overexpression achieved at the highest doxycycline concentrations in these experiments (between 70- and 500-fold, depending on the cell line; S9 Fig) are much larger than those observed in tumors (between 1.5- and 7-fold, Fig 2E). We hypothesized that small changes in SRRM4 expression would have measurable effects on proliferation over a longer period of time, such as during tumor evolution, which occurs on a timescale of years instead of days. In this scenario, cells with lower SRRM4 expression would have a proliferative advantage and would be selected for over time, eventually coming to dominate the population.

To experimentally mimic the competition between cells with high and low SRRM4 expression during tumor evolution, we performed a coculture experiment where MDA-MB-231 cells with inducible expression of either WT or DM SRRM4 were mixed at a 1:1 ratio. The cocultures were then passaged for several weeks in the presence of varying levels of doxycycline, such that the lowest concentration (31.3 ng/mL) achieved a similar level of SRRM4 induction as the change in SRRM4 expression observed between tumors and healthy tissues (2.6-fold, S10A Fig). Samples of the cocultures were taken weekly, and the proportion of each cell line in the population was monitored by PCR using primers flanking the deleted region of SRRM4, allowing cells expressing WT or DM SRRM4 to be differentiated by the size of the PCR product. Compared to the ratio of WT to DM cells at Day 1, we observed a dose- and time-dependent decrease in the WT/DM ratio that was significant after 2 wk of coculture at all concentrations tested (Fig 5E and 5F). Similar but even stronger effects were seen in HCT116 (S11 Fig), although the levels of SRRM4 induction were higher in this cell line with the same concentrations of doxycycline (S10B Fig). Taken together, these results strongly suggest that the changes in SRRM4 expression observed between tumors and normal tissues are sufficient to provide a growth advantage to tumor cells with lower SRRM4 expression.

To further validate the finding that SRRM4 expression affects tumor growth, we performed a mouse xenograft experiment using the MDA-MB-231 cells with inducible SRRM4 expression (WT or DM). Mammary fat pads of athymic female nude mice were implanted orthotopically with the inducible breast cancer cells, and tumors were allowed to grow to 60 and 80

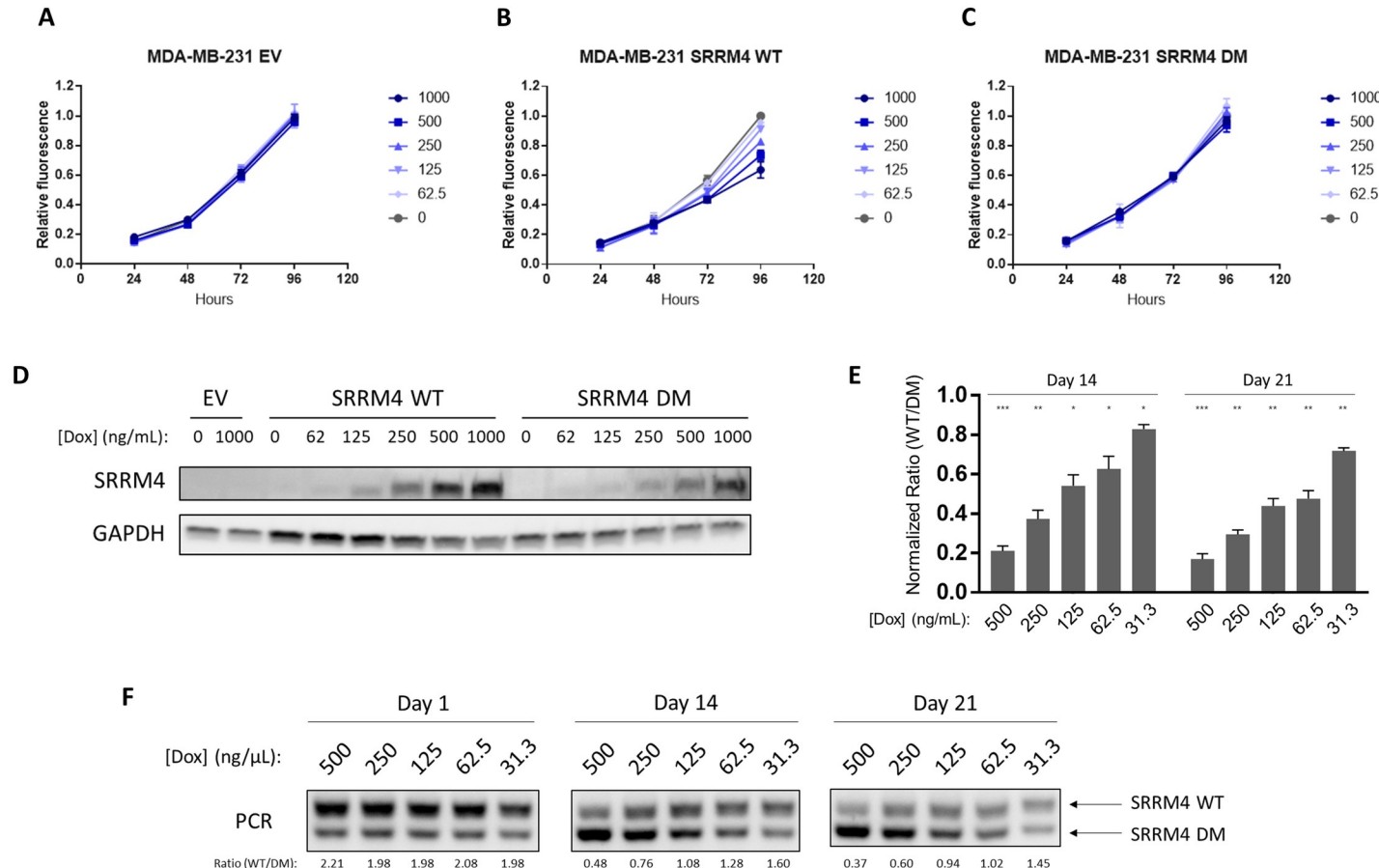

**Fig 5. SRRM4 expression inhibits cancer cell proliferation.** Growth curves of MDA-MB-231 transduced with (A) EV, (B) WT SRRM4, or (C) DM SRRM4, treated with indicated concentrations of doxycycline (ng/mL). Error bars represent SEM of 2–3 independent experiments. (D) Western blot of SRRM4 expression in the same cell lines after 24 h induction with doxycycline at the indicated concentrations. Results are representative of 2 independent experiments. (E) Ratios of WT/DM SRRM4 in MDA-MB-231 cocultures show a dose- and time-dependent decrease (normalized to Day 1) that is significant at every doxycycline concentration tested (unpaired $t$ test using Holm–Sidak method to correct for multiple comparisons; * $p < 0.05$, ** $p < 0.01$, *** $p < 0.001$). Results shown represent the mean of 3 independent experiments, and error bars represent standard deviation. (F) Representative PCR showing the decreased WT/DM ratios of MDA-MB-231 cocultures over time. WT/DM ratios quantified by densitometry are shown below each sample. The apparent WT/DM ratio at Day 1 is approximately 2:1, due to the formation of a chimeric PCR product of intermediate size close to the WT band. However, the ratios quantified at Day 14 and Day 21 decrease with respect to Day 1 in both a dose- and time-dependent manner, indicating that the DM cells outcompete the WT. Data underlying Fig 5A–5C and 5E can be found in S1 Data figures. DM, deletion mutant; EV, empty vector; SEM, standard error of the mean; SRRM4, Serine/Arginine Repetitive Matrix 4; WT, wild-type.

mm$^3$ before induction with 2 mg/mL doxycycline in drinking water (Fig 6A). In agreement with the in vitro experiments, induction of WT-SRRM4 tumors with doxycycline significantly decreased the rate of tumor growth compared with uninduced tumors (Fig 6B). In contrast, induction of DM-SRRM4 had no significant effect on tumor growth (Fig 6C). Overall, these results validate the hypothesis that SRRM4 splicing activity can influence tumor growth in vitro and in vivo.

## SRRM4 expression induces neuron-like expression and splicing patterns in cancer cells

To evaluate the functional consequences of altered SRRM4 expression in the above cancer cell lines, we performed RNA-seq after 24 h induction of SRRM4 expression. Using the inactive

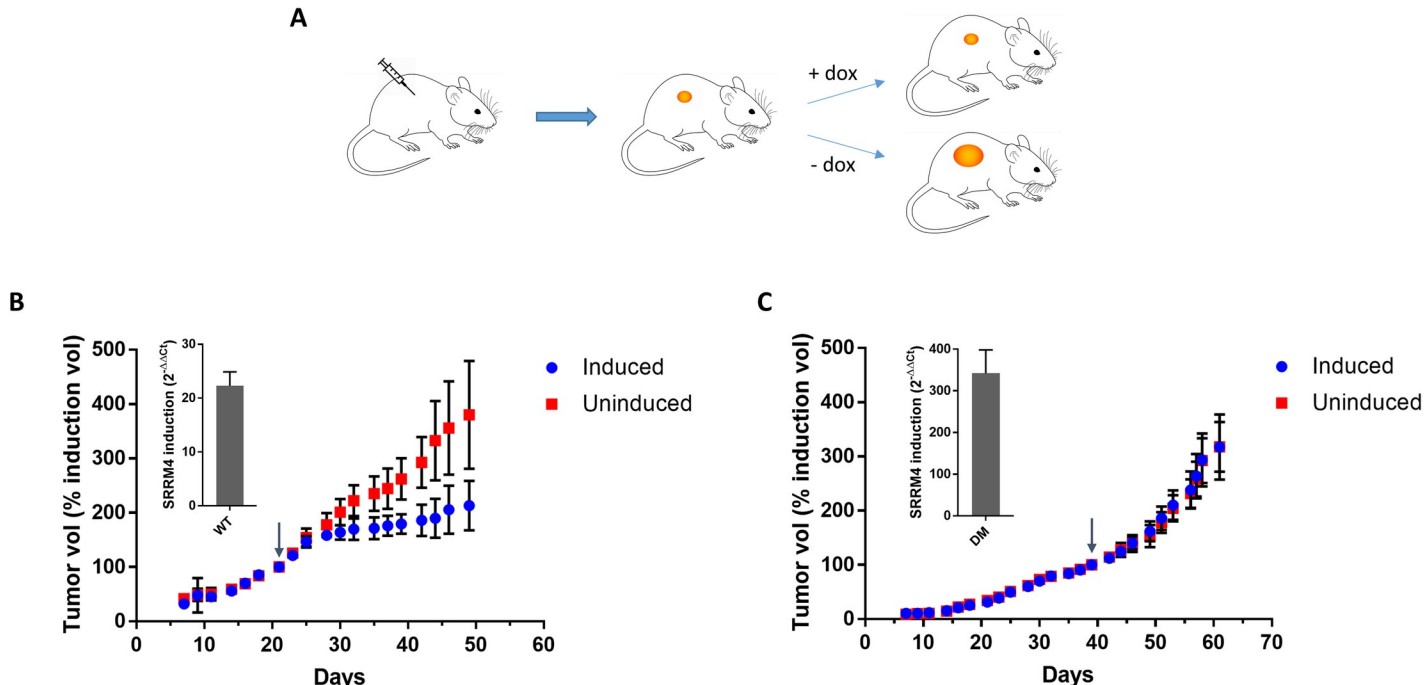

**Fig 6. SRRM4 induction inhibits tumor growth in vivo.** (A) Schematic overview of mouse experiment. Mice were implanted with MDA-MB-231 cells with inducible SRRM4 expression (WT or DM), and tumors were allowed to grow to a certain size before induction with doxycycline in drinking water (2 mg/mL). (B) WT SRRM4 tumors grew significantly more slowly in mice receiving doxycycline than uninduced mice. Tumor volume was normalized to tumor volume at the time of induction (arrow). Error bars represent standard deviation, $n$ = 6 per group. Inset: SRRM4 overexpression in induced tumors relative to uninduced tumors was quantified by qPCR using the ΔΔCt method, with GAPDH as internal control. Error bars represent SEM of 2 experimental replicates. (C) Tumors expressing the inactive SRRM4 mutant (DM) showed no difference in growth rate with or without doxycycline induction. Tumor volume was normalized to tumor volume at the time of induction (arrow). Error bars represent standard deviation, $n$ = 6 per group. Inset: SRRM4 overexpression in induced tumors relative to uninduced tumors was quantified by qPCR using the ΔΔCt method, with GAPDH as internal control. Error bars represent SEM of 2 experimental replicates. Data underlying Fig 6B and 6C can be found in S1 Data figures. DM, deletion mutant; qPCR, quantitative PCR; SEM, standard error of the mean; SRRM4, Serine/Arginine Repetitive Matrix 4; WT, wild-type.

mutant as a reference, we detected increased inclusion (ΔPSI ≥ 25) of between 271 and 376 exons, depending on the cell line (S6 Data). In total, we found 708 exons with increased inclusion in at least one of the 6 cell lines and 131 that were shared between all 6 cell lines (Fig 7A). Of these, a majority (70.2%) were microexons (Fig 7A inset). Although these 131 shared exons generally display low inclusion in normal nonneural tissues compared with neural (Fig 7B), enrichment analysis confirmed that they were significantly overrepresented among the exons decreasing in all 9 cancer types ($p < 0.05$, 1-sided Fisher Exact test), further validating the silencing of SRRM4 target exons in tumors.

Six microexons were chosen for further validation, based on their decreased inclusion in multiple TCGA tumor types (S12–S17 Figs, panel a) and their differential splicing between WT and DM cells in all 6 cell lines (S12–S17 Figs, panel b). In all cases, PCR with primers flanking the microexon confirmed the differential splicing observed by RNA-seq for all cell lines (S12–S17 Figs, panel c). We also found that each exon had significant negative correlations with MI gene signature in a majority of tumor types (S2 Table), suggesting that downregulation of these exons might contribute to the growth advantage obtained by tumors with decreased SRRM4.

In addition to splicing changes, we also monitored changes in gene expression in the 6 cell lines after 24 h induction of WT or DM SRRM4 expression, and two of the cell lines (MDA-MB-231 and SH-SY5Y) after 7 d of induction. Compared to the DM control, WT

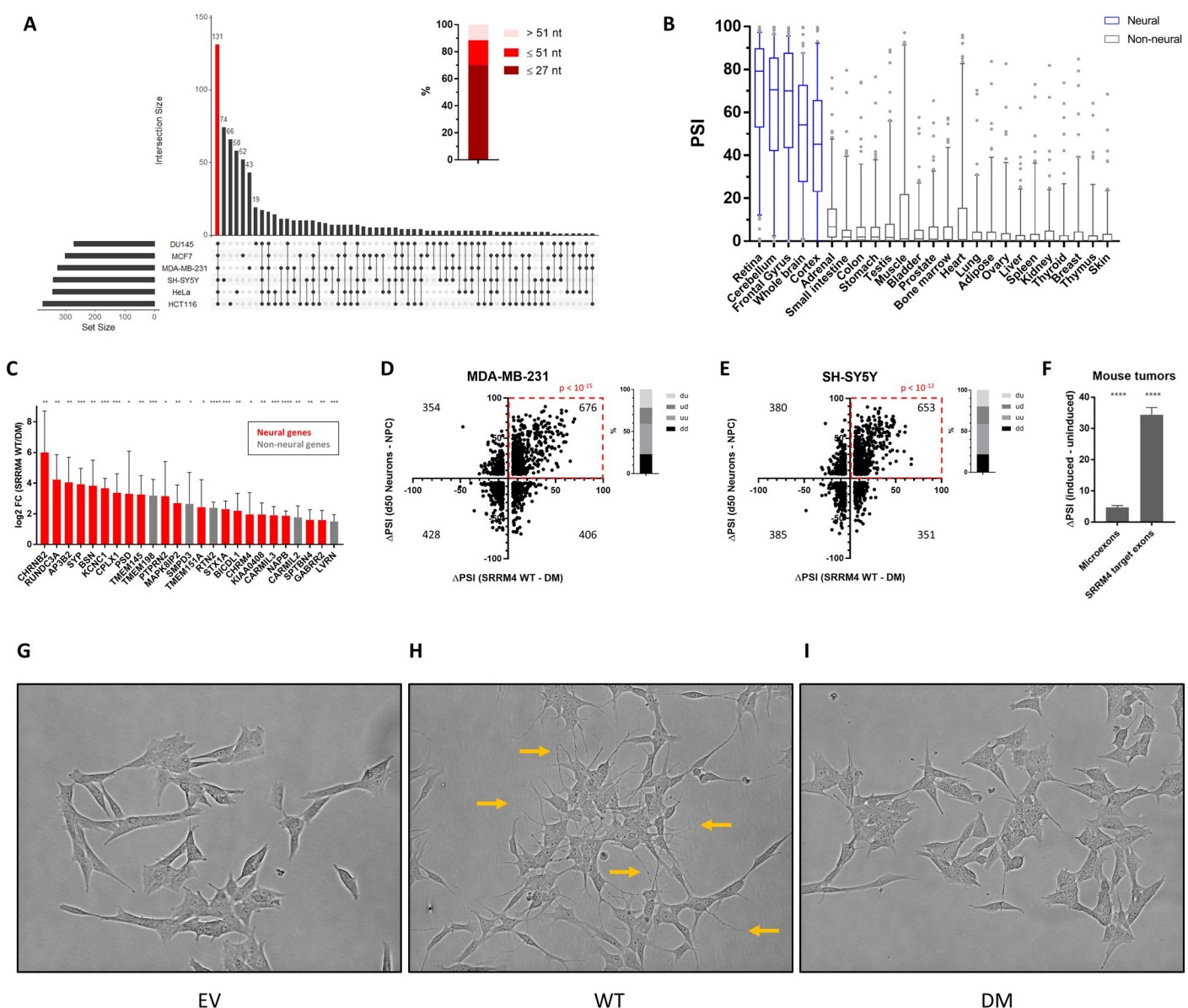

**Fig 7. SRRM4 expression leads to neural-like expression and splicing patterns and morphological changes in cancer cells.** (A) Exons with increased inclusion in all 6 cell lines after SRRM4 induction ($\Delta$PSI $\geq$ 25). Inset: nucleotide lengths of the 131 shared exons, demonstrating a majority are microexons. The cell lines comprising each intersection set are indicated below each bar. Figure was generated using UpSet package in R. (B) Box plots of the PSIs of the 131 SRRM4-regulated exons shared between the 6 cell lines across tissue types (from VastDB). Boxes expand from the first to the third quartile, with the center values indicating the median, and whiskers extend from 5 to 95 percentile. (C) Genes with increased expression in all 6 cell lines after SRRM4 induction (fold change $\geq$2 compared to DM control). Bars represent the mean fold change values, with error bars representing 95% CI. Red bars are genes with known neuronal function (GO: nervous system development) and/or neural-specific expression patterns. Fold change values shown are significantly different from 0 (1-sample $t$ test, 2-tailed; *$p < 0.05$, ** $p < 0.01$, *** $p < 0.001$, **** $p < 0.0001$). (D, E) Overlap between changes in exon inclusion with neuronal differentiation dataset after 7 d of SRRM4 induction in (D) MDA-MB-231 or (E) SH-SY5Y ($p$-values shown from binomial test). (F) Changes in inclusion of microexons ($\leq$27 nt) and SRRM4 target exons between the doxycycline-induced and uninduced mouse tumors. Bars represent the mean $\Delta$PSI of the sets of exons, with error bars representing 95% CI. The mean $\Delta$PSI for both sets of exons was significantly different than 0 (1-sample $t$ test, 2-tailed; **** $p < 0.0001$). (G, I) SH-SY5Y cells were transduced with (G) EV, (H) WT SRRM4, or (I) DM SRRM4. Cell images were taken after 7 d induction with 1 µg/mL doxycycline. WT SRRM4-expressing cells develop numerous membrane projections (orange arrows). Results shown are representative of 3 independent experiments. Data underlying Fig 7A–7F can be found in S1 Data figures. DM, deletion mutant; EV, empty vector; GO, gene ontology; nt, nucleotide; PSI, percent spliced in; SRRM4, Serine/Arginine Repetitive Matrix 4; WT, wild-type.

SRRM4-expressing cells showed an up-regulation of a large number of neuronal-specific genes. Among the genes found up-regulated ($\geq$2-fold) in all 6 cell lines after 24 h were those with functions related to neurotransmitter release (AP3B2, BSN, CPLX1, STX1A, and SYP), neurotransmitter receptors (CHRM4, CHRNB2, and GABRR2), and other genes with neural-specific expression patterns [21] (CARMIL3, MAPK8IP2, NAPB, TMEM145, TMEM151A, and RUNDC3A) (Fig 7C, S7 Data). We further performed a global comparison of expression changes in the SRRM4-expressing cancer cells (WT versus DM) with a published dataset of human embryonic stem cells (ESCs) differentiated into neurons in vitro (mature neurons [50 d post-differentiation] versus proliferating neuronal progenitor cells [29]). This comparison revealed a significant positive association in all cell lines at both 24 h and 7 d induction time points ($p < 0.005$, binomial test; S8 Data), where a majority of genes with increased expression during neuronal differentiation were also found to increase with WT SRRM4 induction. A similar positive association was also seen for changes in exon inclusion between datasets in all cell lines ($p < 10^{-8}$, binomial test; Fig 7D and 7E, S9 Data), in agreement with a previous study comparing splicing differences between neural and nonneural tissues with those seen in the brains of SRRM4 KO mice [5]. The similarity between changes observed in the neuronal differentiation dataset and the cell lines with SRRM4 overexpression further supports the hypothesis that SRRM4 promotes differentiated neuron-like expression and splicing patterns in cancer cells.

We additionally performed RNA-seq on a small sample of tumors from the mouse xenograft experiment, comprising 1 uninduced tumor (U19) and 2 induced tumors (I2 and I14). As expected, both induced tumors had significantly increased inclusion of all microexons, and SRRM4 target exons specifically, compared to the uninduced tumor (Fig 7F), as well as up-regulated expression of neural genes similar to that observed in the cell lines in vitro (S10 Data). Likewise, the differences in both gene expression and exon inclusion between induced and uninduced tumors also showed a highly significant positive association with changes seen in the neuronal differentiation dataset ($p < 10^{-12}$, binomial test; S11 and S12 Data). These results confirm that the differentiation-promoting program induced by SRRM4 has antiproliferative effects on cancer cells in vivo.

Consistent with the observed changes in gene expression and exon inclusion, we also observed striking morphological changes in SH-SY5Y upon SRRM4 overexpression, where cells developed numerous membrane projections after 1 wk of induction (Fig 7G–7I). This morphological change is reminiscent of SH-SY5Y cells that have undergone neuronal differentiation with retinoic acid (RA) and brain-derived neurotrophic factor (BDNF) treatment [30–32]. Indeed, the expression changes observed in both SH-SY5Y and MDA-MB-231 after 7 d induction displayed a strong positive association with a published dataset of in vitro differentiation of SH-SY5Y cells using RA/BDNF [33] ($p < 10^{-7}$, binomial test; S13 Data). While we did not observe such morphological changes in the other cancer cell lines, the consistent up-regulation of neural expression and splicing patterns across lines suggests that these cells are shifting toward a more neural differentiation–like state but may not be primed for developing morphological changes, at least after 7 d. However, a recent study reported an increase in cell projections and decreased cell body size in DU145 cells after constitutive long-term SRRM4 overexpression [34], implying that such changes can occur over longer periods of time.

Overall, the above results demonstrate that expression of SRRM4 in cancer cells leads to an up-regulation of microexon inclusion and neuron-like expression and splicing patterns, which occurs concomitantly with a decrease in cell proliferation. Taken together, these findings provide a mechanistic explanation for the observed decrease of SRRM4 expression and microexon inclusion in cancer.

## Discussion

In this study, we demonstrate that tumors exhibit down-regulated expression of the microexon SF SRRM4 and inclusion of its target exons, with remarkably high consistency across tissues. Our results suggest that this down-regulation provides tumors with a growth advantage, as decreased SRRM4 expression in tumors is correlated with an increase in mitotic gene expression, and up-regulation of SRRM4 in cancer cell lines dose-dependently decreases proliferation. This antiproliferative activity is also linked to the induction of neuronal differentiation–related genes and splicing patterns, in accordance with the known function of SRRM4 in promoting neuronal differentiation. Proliferation and differentiation are widely considered to be distinct and antagonistic cell states, where terminal differentiation is normally accompanied by exit from the cell cycle and loss of proliferative capacity, and conversely, cancer cells evade pro-differentiation programs to promote their proliferation and self-renewing abilities [29,35–38]. We therefore surmise that silencing of SRRM4 provides a selective advantage to cancer cells by shifting the cell state away from differentiation and toward proliferation.

SRRM4 is largely considered to be a neural-specific SF, and as such its basal expression and inclusion of its target exons are very low outside of the brain. Therefore, our finding that SRRM4 is further silenced in cancers of nonneural origin suggests that low levels of SRRM4 outside of the brain do have a physiologic role as proliferation brakes. Although the effect of down-regulating an already lowly expressed gene might be expected to be minimal, any small proliferative advantage would be amplified over time, eventually dominating the cell population. Moreover, despite the fact that around two-thirds of SRRM4-regulated exons are not included outside of the brain, we observe that a majority of these exons that change in cancer are decreased in tumors, implying that the one-third of SRRM4 exons that are nonzero outside of the brain are sufficient to promote this antiproliferative effect.

While it is well documented that differentiating neurons stop dividing and enter a postmitotic, terminally differentiated state upon reaching maturity, the molecular mechanisms mediating this proliferation arrest are incompletely understood. Interestingly, mature neurons cannot be induced to reenter the cell cycle without resulting in cell death, and accordingly, terminally differentiated neurons are unable to form tumors without first undergoing dedifferentiation [39–43]. It is noteworthy that increased SRRM4 expression and microexon inclusion occur concomitantly with a cessation of cell division during differentiation [5,44,45]. We therefore speculate that SRRM4, in addition to suppressing growth in nonneural tissues, may be involved in mediating proliferation arrest during neuronal differentiation; however, this possibility remains to be explored in future studies.

Unlike AS events that add or remove entire functional protein domains, microexons are thought of as modulators that "fine-tune" protein activity and protein–protein interaction interfaces [1]. As such, while each individual microexon may have moderate effects on individual processes, modulation of the entire program by the central regulator SRRM4 is a more efficient driver of differentiation by affecting many processes at once. Accordingly, although we find that SRRM4-regulated exons on the whole are down-regulated in tumors across all tissues, we do not find any individual SRRM4 exon that changes significantly in every tissue, supporting the idea that the antiproliferative phenotype is likely due to a combined effect of multiple exons that promote neural differentiation. However, we cannot rule out the existence of exons that did not satisfy our ΔPSI cutoff in all tissues but still have biological significance. Differences across tissues for individual exons could also be due to varying expression levels of the SRRM4 target genes or additional layers of regulation by other SFs.

Other SFs aside from SRRM4 have been shown to be involved in the regulation of microexon splicing, including SRRM3, SRSF11, RBFOX, and PTBP1 [19,22,46]. While SRRM4 is the

only one of these factors that changes significantly and in the same direction across all tumor types included in our study, others also display significant expression changes in specific tumor types (S1 Data). Therefore, we cannot rule out their potential involvement in the regulation of individual microexons in certain tissues. Interestingly, our results suggest that SRRM3 is not involved, despite being thought to regulate similar targets as SRRM4, as its expression is increased in a majority of tumor types (Fig 2C). This supports the idea of differing roles for SRRM3 and SRRM4 in vivo, as has been previously proposed by others [47]. In addition to the factors mentioned, it is likely that other SFs identified here as having consistent changes across all tumor types (SNRPB, RNPS1, RBM34, ELAVL1, and RALYL; Fig 2E and 2G) could contribute to tumor-associated splicing events in the analyzed datasets. Due to the high degree of splicing dysregulation in cancer generally, as well as the complex crosstalk between SFs which often regulate each other's activity [14,48], it is very difficult to attribute observed splicing changes in cancer to one specific factor. Nevertheless, our findings of consistently decreased SRRM4 expression across tumor types (Fig 2A), coupled with the correlation of SRRM4 expression with inclusion of its target exons in tumors (S4A Fig), and their significant overlap with exons increasing after SRRM4 overexpression in different cancer cell lines, provide strong evidence that SRRM4 dysregulation contributes substantially to the observed splicing changes in cancer.

Among the microexons identified with differential splicing in both the tumors and cell lines, several have reported functions in proliferation and tumorigenesis. For example, TEAD2 is a transcription factor that regulates gene expression downstream of the hippo signaling pathway, a key regulator of cell proliferation that is frequently dysregulated in cancer [49]. The inclusion of a 12-nucleotide exon in TEAD2 (event ID: HsaEX0064372) is one of the most consistently decreased microexons in tumors (6/9 tissue types in our dataset; S12A Fig) and is SRRM4 regulated, as evidenced by its up-regulation in all 6 cancer cell lines after SRRM4 overexpression (S12B and S12C Fig). Although the function of this particular microexon has not been studied, inclusion of a paralogous microexon in TEAD1 has been shown to disrupt DNA binding and thereby inhibit cell proliferation in hepatocytes [50]. Other examples of cancer-related genes with differentially spliced microexons include SPTAN1 (S13 Fig), a ubiquitously expressed cytoskeletal protein which has recently been shown to affect tumor growth by enhancing cell proliferation and migration [51]; DOCK7 (S14 Fig), which regulates the RAGE-Cdc42 pathway that promotes cancer progression [52] and glioblastoma cell invasion [53]; and PSAP (S15 Fig), which has been implicated in the tumorigenesis of glioblastoma [54], prostate [55,56], and breast cancer [57]. In support of possible antiproliferative roles for these exons, we find significant negative correlations between their inclusion and MI across multiple tumor types (S2 Table). Thus, microexon splicing changes affecting the activity of multiple genes with known roles in cancer likely contribute to the antiproliferative phenotype of SRRM4. Our ongoing work aims to identify the roles of individual microexons in proliferation and tumorigenesis.

Prior studies of SRRM4 in cancer to date have focused exclusively on a rare class of tumors known as neuroendocrine (NE) tumors (0.5% to 2% of malignancies in adults), due to their shared properties with neural or NE tissues [58]. A series of recent studies have suggested that SRRM4 may play a role in the progression of small-cell lung cancer and NE prostate cancer by inactivating the RE1-silencing transcription factor (REST), in turn promoting expression of neuronal genes [34,59–65]. This program appears to promote transdifferentiation and is associated with poor prognosis in these tumors [59,62,66]. In fact, it has been suggested that SRRM4 mediates increased SOX2 expression, driving prostate cancer cells to a pluripotent phenotype and favoring tumor growth [34]. While we show that SRRM4 overexpression inhibits cell proliferation in all cancer cell lines tested (including the prostate cancer cell line

DU145), in our experiments, SRRM4 did not induce SOX2 expression, which could explain the apparent contradiction. NE tumors with SOX2 expression are therefore likely the exception to the here-described selection for silencing of SRRM4 in cancer. Notably, our findings suggest that prostate cancers and lung cancers on the whole follow the opposite trend as has been shown by others for their NE subtypes [59,64].

Previous studies of AS alterations in cancer have not specifically taken microexons into account, due to the technical challenges presented by their detection as well as a long-standing underappreciation of their biological relevance. As such, the finding that SRRM4 and its microexon splicing program are silenced in tumors contributes an important new element to the overall picture of splicing dysregulation in cancer. Furthermore, the unusual discovery of such a highly consistent mechanism spanning diverse tissue types may provide novel opportunities for therapeutic intervention in a wide range of cancers. With the advent of splice-switching antisense oligonucleotides that are now being used therapeutically to affect exon inclusion levels [67], this knowledge could guide the development of new strategies for cancer treatment.

## Methods

### Statistical information

Differential expression/methylation of SFs and differential exon inclusion were determined by 2-tailed unpaired Wilcoxon–Mann–Whitney test, using the Benjamini–Hochberg method to correct for multiple testing. The numbers of samples per group are indicated in S1 Table, and $p$- and q-values are reported in S1–S3 Data. For the distributions shown in S4 Fig, significance was determined by 1-tailed randomization test of the median correlation of SRRM4-regulated events among the corresponding background. For the GO analysis, we used the Cytoscape plug-in in ClueGO (v2.5.4) [24]. From the list of genes containing down-regulated exons, we applied a 2-sided hypergeometric test to identify enriched GO terms (between the hierarchical GO levels 3 and 5), with Bonferroni step down correction. All terms containing a minimum of 10 genes and constituting at least 1% of the term were included. Binomial tests for associations between changes in different RNA-seq datasets used cutoffs of $\geq$2-fold change in gene expression or $\Delta$PSI $\geq$ 5 for exon inclusion, and $p$-values were calculated using the binom.test function in R version 3.4.4. Exact gene expression/exon inclusion values, sample sizes, and $p$-values for each binomial test performed can be found in S8, S9, and S11–S13 Data.

### TCGA molecular tumor data

We retrieved publicly available and preprocessed RNA-seq gene expression from FireBrowse and methylation quantification from MethHC [68]. Raw sequencing data in fastq format were retrieved from the GDC legacy archive after obtaining the necessary permissions from dbgap. Tumor grades and mutation status of cancer driver genes were obtained from FireBrowse. Tumor purity estimates were retrieved from [28].

### Detection of differentially included exons in tumors

To determine differential microexon splicing, we analyzed raw RNA-seq files using VAST-TOOLS v2.0, a computational pipeline designed specifically for the quantification of microexon inclusion levels [1]. We ran VAST-TOOLS on 6,908 samples (S1 Table). Using VAST-TOOLS align, we first mapped the reads to the human reference genome hg38 and to quantify PSI levels for each exon and sample. We used VAST-TOOLS tidy to remove exon quantifications with very low quality. We additionally required that PSI levels were determined in at

least 20 of both tumor and matched healthy control samples. We then determined differentially spliced exons by testing for differences in the PSI distributions between tumor and control samples using the Wilcoxon–Mann–Whitney test. We performed FDR correction and considered exons significantly spliced for q-values < 0.01 and an absolute PSI difference between the 2 distributions ≥5.

### Definition of splicing factor list

The list of 202 SFs used in this study was derived from a published list of known SFs in mice [69] for which 1:1 orthologs could be identified in humans, plus 6 manual additions of human genes with evidence of SF activity or family members thereof: ARGLU1 [70], PCBP2 [71], PRPF18 [72], SNIP1 [73], SRRM3 [22], and SFRS2 [74]. The complete list can be found in S1 Data.

### Definition of SRRM4 target exon list

For this study, we generated independent biological replicates for the HEK293 line overexpressing WT SRRM4 as well as a control line overexpressing green fluorescent protein [22] and performed RNA-seq (Sequence Read Archive accession number PRJNA474911). We used VAST-TOOLS to quantify AS in these samples and the corresponding matching samples from a previous study [22] (SRX4193731 and SRX4193721). A list of 314 exons with increased inclusion, i.e., ΔPSI between WT SRRM4 and control ≥25 in both replicates, was defined as SRRM4 targets and subsequently used in further analyses.

### Correlation between SRRM4 activity and target exon PSI levels

SRRM4 target exon PSI was correlated against the expression and methylation of a set of background genes, in order to determine the significance of SRRM4 correlation by a randomization test. Three different background sets were included: (1) a random set of genes (5,000 in expression background, 500 in methylation background); (2) 425 genes that are GO related to splicing; and (3) 233 genes that are reported to regulate microexons [46].

### Plasmids

The cloning of human WT and DM (ΔeMIC) SRRM4 was described previously [22]. SRRM4 was amplified by PCR using attB-containing primers (S3 Table) and reacted with the gateway donor vector pDONR223 by using BP clonase II (11789020, Thermo Fisher Scientific, Madrid, Spain). The resulting entry clones containing SRRM4 WT and DM were reacted with the gateway destination vector pCW57.1 (a gift from David Root, Addgene plasmid #41393) using LR clonase II (11791020, Thermo Fisher Scientific).

### Cell lines and lentivirus production

HEK293 cells were purchased from ATCC (CRL-1573, Manassas, Virginia, United States of America). DU145 were a gift from Luciano Di Croce (CRG). All other cell lines were obtained from the CRG cell line repository. The sex of each cell line is as follows: DU145 Male; HeLa Female; HEK293 Female; MCF7 Female; MDA-MB-231 Female; SH-SY5Y Female; and HCT116 Male. All cell lines were cultured at 37°C and 5% CO2 in DMEM (H3BE12-604F/U1, Lonza Group AG [Cultek S.L.U, Madrid, Spain]) supplemented with 10% tetracycline-free FBS (631106, Clontech, [Conda Laboratories, Torrejon de Ardoz, Spain]) and 1% penicillin/streptomycin (15140122, Life Technologies S.A., [Fisher Scientific, Madrid, Spain]).

To generate lentivirus, low passage HEK293T cells were plated in 6-well plates at 800,000 cells/well in 3 mL media. The following day, cells were co-transfected with 0.9 μg pCW57.1 (EV, SRRM4 WT, or SRRM4 DM), 0.6 μg psPAX2 (a gift from Didier Trono, Addgene plasmid #12260), and 0.3 μg pMD2.G (a gift from Didier Trono, Addgene plasmid #12259) per well using Lipofectamine 2000 transfection reagent (11668027, Thermo Fisher Scientific) according to the manufacturer's instructions. After 48 to 72 h, viral supernatant was harvested, filtered through a 0.45 um PVDF filter (SLHV033RS, Merck Chemicals and Life Science S.A, Madrid, Spain), and transferred to the target cell line growing in a 6-well plate at approximately 70% confluence. After 48 h, the target cells were trypsinized and replated in 6-cm dishes in media containing 2 μg/mL Puromycin (P8833, Sigma-Aldrich Quimica S.A., Madrid, Spain). Selection media was changed every 48 to 72 h for a minimum of 1 wk before cell lines were frozen in 50% FBS and 10% DMSO and stored in liquid nitrogen for later use. The generation of HEK293 with inducible WT SRRM4 expression is described elsewhere [22].

## RNA extraction and sequencing

Cell extracts were harvested by direct addition of QIAzol lysis reagent (79306, Qiagen [WERFEN ESPAÑA, S.A.U. L'Hospitalet de Llobregat, Barcelona]) to the culture plate and incubation for 5 min at room temperature (RT) followed by collection in 1.5 mL eppendorf tubes. RNA was isolated using the miRNeasy mini kit (217004, Qiagen [WERFEN ESPAÑA, S.A.U.]) according to the manufacturer's instructions. For isolation of RNA from mouse tumor tissue, tumors frozen in RNAlater stabilization solution (AM7020, Thermo Fisher Scientific) were thawed in QIAzol and disrupted using a Polytron PT1200E homogenizer (VWR [Llinars de Vallés, Spain]), before RNA extraction using the RNeasy Plus Mini Kit (74134, Qiagen [WERFEN ESPAÑA, S.A.U.]). Purified RNA was stored at −80˚C before sequencing. RNA quality control, library preparation, and sequencing were performed by the CRG Genomics Unit. RNA concentration and quality were assessed by nanodrop and bioanalyzer. An average of 90 million 125-nucleotide paired-end reads were generated for each sample. Raw sequencing data were submitted to the Sequence Read Archive (accession number PRJNA551123).

## RT-PCR

RNA was reverse transcribed into cDNA using SuperScript II Reverse Transcriptase (18064, Thermo Fisher Scientific) with random hexamer primers (SO142, Thermo Fisher Scientific) according to the manufacturer's instructions. PCRs for quantification of microexon inclusion (shown in S12–S17 Figs, panel c) and competition growth assay (shown in Fig 5F and S11 Fig) were performed using Q5 Hot Start High-Fidelity 2X Master Mix (M0494, New England Biolabs [Ipswitch, Massachusetts, USA]) according to the manufacturer's instructions. Microexon PCR products were separated on 3% agarose gel with TBE buffer, while WT/DM SRRM4 PCR products were separated on 1.2% agarose gel with TAE buffer. PCR primer sequences are shown in S3 Table.

## qPCR

Quantitative PCR (qPCR) reactions were carried out with the Roche Light Cycler 480 system, using Power SYBR-Green PCR Master Mix (4367659, Thermo Fisher Scientific) according to the manufacturer's instructions. Expression quantification data, obtained from the different samples and treatments, are expressed in terms of cycle threshold (Ct). The *GAPDH* gene was used as endogenous reference control. Ct values were averaged for each in-plate technical triplicate. The averaged Ct was normalized as difference in Ct values (ΔCt) between the analyzed gene and each reference gene in each sample in the analysis. The ΔCt values of each sample

were then normalized to the ΔCt values of the control (ΔΔCt). The variation was reported as fold change ($2^{-\Delta\Delta Ct}$). qPCR primer sequences are shown in S3 Table.

## Resazurin growth assays

Resazurin sodium salt (R7017, Sigma-Aldrich Quimica S.A.) stock was prepared at 0.2 mg/mL in PBS and sterilized by filtration through a 0.22 um PVDF filter (SLGP033RS, Merck Chemicals and Life Science S.A). Cells were seeded in 4 flat-bottom 96-well plates (353072, Corning B.V. [Fisher Scientific, Madrid, Spain]) at a density of 1,000 cells/well in 200 μL of tetracycline-free media, containing 2-fold serial dilutions of doxycycline (D9891-1G, Sigma-Aldrich Quimica S.A.) (1000–62.5 ng/mL final concentration) with each condition in triplicate. Every 24 h, resazurin was added to 1 plate (10 μL/well), and fluorescence intensity was measured after 5 h using an Infinite M-Plex plate reader from Tecan (Ibérica Instrumentación S.L, Barcelona, Spain) (ex = 560 nm, em = 590 nm). Within each experiment, values were normalized to the mean fluorescence of uninduced cells at the 96 h time point. Three replicate experiments were performed for each cell line. The results were graphed using GraphPad Prism v. 7.04 and are presented as normalized means with error bars representing the standard error of the means.

## Competition growth assays

Cells with inducible expression of WT or inactive DM SRRM4 were seeded at 100k cells/well each into 6-well plates containing 2 mL of media supplemented with varying concentrations of doxycycline (500, 250, 125, 63, 31, or 0 ng/mL) on Day 0. The cocultures were split at Day 1 and a sample taken for RNA extraction, while the remaining cells were replated in media with the same concentrations of doxycycline for continued culture. The cocultures were split every 2 to 3 days into fresh media with doxycycline and were harvested every week for RNA extraction for up to 3 wk. RNA extraction, cDNA preparation, and PCR were performed as described in the reverse transcription PCR (RT-PCR) section. The signal intensity of WT and DM bands was quantified by densitometry using ImageJ software. To account for the apparent difference in PCR efficiency between the 2 bands, WT/DM ratios were normalized to the ratio observed at Day 1 for each doxycycline treatment condition. PCR primer sequences are shown in S3 Table.

## Western blotting

Cell extracts were harvested by direct addition of 2X SDS sample buffer (1610747, Bio-Rad Laboratories, Alcobendas, Spain) containing 100 mM DTT to the culture plate and incubation on ice for 10 min, followed by collection in 1.5 mL eppendorf tubes and denaturation for 5 min at 95˚C. Lysates were subjected to SDS-PAGE in Criterion TGX Stain-Free Precast Gels (567–8124, Bio-Rad Laboratories) and transferred to nitrocellulose membranes (IB301001, Thermo Fisher Scientific) by iBlot Dry Blotting System (Thermo Fisher Scientific). Membranes were blocked in 5% milk in TBS with 0.05% Tween-20 for 1 h at RT before addition of primary antibodies (SRRM4, HPA052783, Sigma-Aldrich Quimica S.A.; GAPDH, sc-47724, Santa Cruz Biotechnology [Dallas, Texas, USA]) diluted 1:1,000 in blocking buffer, followed by overnight incubation at 4˚C. Membranes were washed 3 × 5 min in TBS-T, followed by 1 h incubation at RT in secondary antibodies (Anti-Rabbit: A0545; Anti-Mouse: A6782, Sigma-Aldrich Quimica S.A.) diluted 1:10,000 in blocking buffer, and 3 more 5 min washes with TBS-T. Finally, membranes were developed by the addition of HRP substrate (34096, Thermo Scientific, [Fisher Scientific, Madrid, Spain]) and exposed using a LAS-3000 Imaging System from Fujifilm (Tokyo, Japan).

## Mouse xenograft model

Mouse experiments were performed by Axis Bioservices (Coleraine, Northern Ireland). Female athymic nude mice aged 5 to 7 wk weighing approximately 23 to 30 g were bred in house and housed in individually ventilated cages (IVCs) (up to 5 mice per cage) with individual mice identified by tail mark. Holding conditions were maintained as follows: RT 20 to 24°C, humidity 30% to 70%, and 12 h light/dark cycle. All animals had free access to standard a certified commercial diet and water, and cages were changed once a week with food and water replaced when necessary. Animals were randomly allocated to treatment groups using GraphPad. MDA-MB-231 cells with inducible SRRM4 expression, generated as described above, were cultured in DMEM with 10% tetracycline-free FBS, 1% penicillin/streptomycin, and 1 μg/mL puromycin. Cells were orthotopically implanted into the inguinal mammary fat pad of the mice using a 25G needle ($5 \times 10^6$ cells in Matrigel 1:1, 1 tumor implanted per mouse). When tumors reached approximately 60 to 80 mm³, mice were randomly assigned to treatment groups using GraphPad and treated with or without doxycycline diluted to a concentration of 2 mg/mL in drinking water. Tumor volume was measured 3 times per week using digital calipers, and measurements continued for 4 wk after induction with doxycycline. The length and width of the tumor were measured, and volume was calculated using the following formula: volume = (length × width²)/2. The bodyweight of all mice was measured and recorded 3 times per week, and mice were observed daily for any signs of distress or changes to general condition. After 28 d post-doxycycline induction, animals were humanely killed via carbon dioxide inhalation, and tumors were resected and weighed before being placed into RNAlater stabilization solution and frozen at −80°C for later analysis.

## Ethics statement

All animal protocols were approved by the Axis Bioservices Animal Welfare and Ethical Review Board (project license number PPL2862), and all procedures were carried out under the guidelines of the Animal (Scientific Procedures) Act 1986.

## Supporting information

**S1 Fig. Average ΔPSI (Tumor − Normal) across tissues for each SRRM4 target exon.** Error bars represent the standard error of ΔPSI across the 9 cancer types. Only significantly changing exons (q < 0.01) with ΔPSI ≥ 5 are shown. Data underlying this figure can be found in S1 Data figures. PSI, percent spliced in; SRRM4, Serine/Arginine Repetitive Matrix 4. (TIF)

**S2 Fig. Distribution of ΔPSIs of all differentially spliced exons between normal and tumor samples in the TCGA datasets (related to Fig 3A).** Only significantly changing exons (q < 0.01) with ΔPSI ≥ 5 are shown. Data underlying this figure can be found in S3 Data. PSI, percent spliced in; TCGA, The Cancer Genome Atlas. (TIF)

**S3 Fig. Changes in inclusion level of SRRM4 target exons with respect to exon length.** Blue points are SRRM4 target exons increased in tumors (T) vs. normal tissue (N); red are SRRM4 target exons decreased in tumors; gray are non-SRRM4 targets. Only significantly changing exons (q < 0.01) with ΔPSI ≥ 5 are shown. Vertical dotted line separates microexons (≤27 nt) from non-microexons (>27 nt). Data underlying this figure can be found in S3 Data. nt, nucleotide; PSI, percent spliced in; SRRM4, Serine/Arginine Repetitive Matrix 4. (TIF)

**S4 Fig. Median Spearman correlation of (A) SRRM4 expression and (B) SRRM4 methylation with SRRM4 target exons PSI across TCGA tumor samples.** The significance of the correlation was determined based on a randomization test within a background of 5,000 random genes background, as well as 2 other backgrounds of 425 GO splicing–related genes and 233 microexon-related genes. Data underlying this figure can be found in S1 Data figures. GO, gene ontology; PSI, percent spliced in; SRRM4, Serine/Arginine Repetitive Matrix 4; TCGA, The Cancer Genome Atlas.
(TIF)

**S5 Fig. Histogram of MI correlations for microexons (≤27 nt) vs. non-microexons (> 28 nt).** Only exons with $|\Delta PSI| > 5$ and $q < 0.01$ are shown. Data underlying this figure can be found in S1 Data figures. MI, mitotic index; nt, nucleotide; PSI, percent spliced in.
(TIF)

**S6 Fig. Association of SRRM4 expression with tumor-related features of all TCGA cancer types.** (A) SRRM4 expression across different TNM tumor stages. (B) Scatter plots and Spearman correlations between SRRM4 and 2 tumor purity estimates (ESTIMATE and CPE). (C) Association of SRRM4 expression with the presence or absence of mutations in cancer driver genes. Significance is determined by a 2-tailed unpaired Wilcoxon–Mann–Whitney test: $^*p < 0.05$, $^{**}p < 0.01$, $^{***}p < 0.001$, $^{****}p < 0.0001$. Data underlying this figure can be found in S1 Data figures. SRRM4, Serine/Arginine Repetitive Matrix 4; TCGA, The Cancer Genome Atlas; TNM, tumor/node/metastasis.
(TIF)

**S7 Fig. Growth curves of cell lines with inducible SRRM4 expression.** Cells were transduced with EV, WT SRRM4, or DM SRRM4 and treated with indicated concentrations of doxycycline (ng/mL). All lines were generated by lentiviral transduction (see Methods) except for HEK293, which was generated by a Flp-In system [22]. Results shown are averages of 2–3 independent experiments, and error bars indicate SEM. Data underlying this figure can be found in S1 Data figures. DM, deletion mutant; EV, empty vector; SEM, standard error of the mean; SRRM4, Serine/Arginine Repetitive Matrix 4; WT, wild-type.
(TIF)

**S8 Fig. Western blot of SRRM4 (WT) or DM SRRM4 induction by doxycycline.** Cells were treated for 24 h with the indicated concentrations of doxycycline. Results shown are representative of 2 independent experiments. DM, deletion mutant; SRRM4, Serine/Arginine Repetitive Matrix 4; WT, wild-type.
(TIF)

**S9 Fig. Expression levels of SRRM4 in cancer cell lines after 24 h induction with 1 μg/mL of doxycycline.** SRRM4 overexpression relative to uninduced cells was quantified by qPCR using the ΔΔCt method, with GAPDH as internal control. Error bars represent SEM of 2–3 independent experiments. Data underlying this figure can be found in S1 Data figures. Ct, cycle threshold; qPCR, quantitative PCR; SEM, standard error of the mean; SRRM4, Serine/Arginine Repetitive Matrix 4.
(TIF)

**S10 Fig. Expression levels of SRRM4 in (A) MDA-MB-231 cells and (B) HCT116 cells after 24 h induction with indicated concentrations of doxycycline.** SRRM4 overexpression relative to uninduced cells was quantified by qPCR using the ΔΔCt method, with GAPDH as internal control. Error bars represent standard deviation of 2 independent experiments. Data underlying this figure can be found in S1 Data figures. Ct, cycle threshold; qPCR, quantitative

PCR; SRRM4, Serine/Arginine Repetitive Matrix 4.
(TIF)

**S11 Fig. Dose- and time-dependent decrease in the WT/DM ratio of HCT116 cocultures, as monitored by PCR.** WT/DM ratios quantified by densitometry are shown below each sample. Results shown are representative of 2 experiments. DM, deletion mutant; WT, wild-type.
(TIF)

**S12 Fig. Differential splicing of a 12-nt microexon in TEAD2.** (A) The microexon (VastDB event ID: HsaEX0064372) displays significantly decreased inclusion in 6/9 cancer types (2-tailed unpaired Wilcoxon–Mann–Whitney test, $^*$ $p < 0.05$, $^{**}$ $p < 0.01$, $^{***}$ $p < 0.001$, $^{****}$ $p < 0.0001$). No data are shown for normal liver due to a lack of sufficient read coverage in at least 20 samples. Genomic coordinates shown are from human genome reference build hg38. Data underlying this figure can be found in S1 Data figures. (B) The microexon displays increased inclusion in all 6 cancer cell lines after WT SRRM4 expression, relative to DM SRRM4, as assessed by RNA-seq. Data underlying this figure can be found in S9 Data. (C) PCR validation of the microexon inclusion changes in the 6 cancer cell lines. PSIs quantified by densitometry are shown below each sample. DM, deletion mutant; nt, nucleotide; PSI, percent spliced in; RNA-seq, RNA sequencing; SRRM4, Serine/Arginine Repetitive Matrix 4; WT, wild-type.
(TIF)

**S13 Fig. Differential splicing of a 15-nt microexon in SPTAN1.** (A) The microexon (VastDB event ID: HsaEX0061551) displays significantly decreased inclusion in 8/9 cancer types (2-tailed unpaired Wilcoxon–Mann–Whitney test, $^*$ $p < 0.05$, $^{**}$ $p < 0.01$, $^{***}$ $p < 0.001$, $^{****}$ $p < 0.0001$). Genomic coordinates shown are from human genome reference build hg38. Data underlying this figure can be found in S1 Data figures. (B) The microexon displays increased inclusion in all 6 cancer cell lines after WT SRRM4 expression, relative to DM SRRM4, as assessed by RNA-seq. Data underlying this figure can be found in S9 Data. (C) PCR validation of the microexon inclusion changes in the 6 cancer cell lines. PSIs quantified by densitometry are shown below each sample. DM, deletion mutant; nt, nucleotide; PSI, percent spliced in; RNA-seq, RNA sequencing; SRRM4, Serine/Arginine Repetitive Matrix 4; WT, wild-type.
(TIF)

**S14 Fig. Differential splicing of a 15-nt microexon in DOCK7.** (A) The microexon (VastDB event ID: HsaEX0020638) displays significantly decreased inclusion in 6/9 cancer types (2-tailed unpaired Wilcoxon–Mann–Whitney test, $^*$ $p < 0.05$, $^{**}$ $p < 0.01$, $^{***}$ $p < 0.001$, $^{****}$ $p < 0.0001$). Genomic coordinates shown are from human genome reference build hg38. Data underlying this figure can be found in S1 Data figures. (B) The microexon displays increased inclusion in all 6 cancer cell lines after WT SRRM4 expression, relative to DM SRRM4, as assessed by RNA-seq. Data underlying this figure can be found in S9 Data. (C) PCR validation of the microexon inclusion changes in the 6 cancer cell lines. PSIs quantified by densitometry are shown below each sample. DM, deletion mutant; nt, nucleotide; PSI, percent spliced in; RNA-seq, RNA sequencing; SRRM4, Serine/Arginine Repetitive Matrix 4; WT, wild-type.
(TIF)

**S15 Fig. Differential splicing of a 9-nt microexon in PSAP.** (A) The microexon (VastDB event ID: HsaEX0050522) displays significantly decreased inclusion in 7/9 cancer types (2-tailed unpaired Wilcoxon–Mann–Whitney test, $^*$ $p < 0.05$, $^{**}$ $p < 0.01$, $^{***}$ $p < 0.001$, $^{****}$ $p < 0.0001$). Genomic coordinates shown are from human genome reference build hg38. Data underlying this figure can be found in S1 Data figures. (B) The microexon displays increased

inclusion in all 6 cancer cell lines after WT SRRM4 expression, relative to DM SRRM4, as assessed by RNA-seq. Data underlying this figure can be found in S9 Data. (C) PCR validation of the microexon inclusion changes in the 6 cancer cell lines. PSIs quantified by densitometry are shown below each sample. DM, deletion mutant; nt, nucleotide; PSI, percent spliced in; RNA-seq, RNA sequencing; SRRM4, Serine/Arginine Repetitive Matrix 4; WT, wild-type. (TIF)

**S16 Fig. Differential splicing of a 6 nt-microexon in NSFL1C.** (A) The microexon (VastDB event ID: HsaEX0043901) displays significantly decreased inclusion in 5/9 cancer types (2-tailed unpaired Wilcoxon–Mann–Whitney test, * $p < 0.05$, ** $p < 0.01$, *** $p < 0.001$, **** $p < 0.0001$). Genomic coordinates shown are from human genome reference build hg38. Data underlying this figure can be found in S1 Data figures. (B) The microexon displays increased inclusion in all 6 cancer cell lines after WT SRRM4 expression, relative to DM SRRM4, as assessed by RNA-seq. Data underlying this figure can be found in S9 Data. (C) PCR validation of the microexon inclusion changes in the 6 cancer cell lines. PSIs quantified by densitometry are shown below each sample. DM, deletion mutant; nt, nucleotide; PSI, percent spliced in; RNA-seq, RNA sequencing; SRRM4, Serine/Arginine Repetitive Matrix 4; WT, wild-type. (TIF)

**S17 Fig. Differential splicing of a 9 nt-microexon in MTSS1L.** (A) The microexon (VastDB event ID: HsaEX0040737) displays significantly decreased inclusion in 7/9 cancer types (2-tailed unpaired Wilcoxon–Mann–Whitney test, * $p < 0.05$, ** $p < 0.01$, *** $p < 0.001$, **** $p < 0.0001$). Genomic coordinates shown are from human genome reference build hg38. Data underlying this figure can be found in S1 Data figures. (B) The microexon displays increased inclusion in all 6 cancer cell lines after WT SRRM4 expression, relative to DM SRRM4, as assessed by RNA-seq. Data underlying this figure can be found in S9 Data. (C) PCR validation of the microexon inclusion changes in the 6 cancer cell lines. PSIs quantified by densitometry are shown below each sample. DM, deletion mutant; nt, nucleotide; PSI, percent spliced in; RNA-seq, RNA sequencing; SRRM4, Serine/Arginine Repetitive Matrix 4; WT, wild-type. (TIF)

**S1 Table. Tissues and number of PT and normal solid tissue samples analyzed from TCGA.** PT, primary tumor; TCGA, The Cancer Genome Atlas. (XLSX)

**S2 Table. Spearman correlations of exon PSI vs. MI gene signature across tumor samples.** Cells highlighted in red indicate negative correlations, and blue indicate positive. Only significant correlations are shown ($p < 0.05$). NA = not available (gray); ns = not significant (white). MI, mitotic index; PSI, percent spliced in. (XLSX)

**S3 Table. Primer sequences for cloning, RT-PCR, and qPCR experiments.** qPCR, quantitative PCR; RT-PCR, reverse transcription PCR. (XLSX)

**S1 Data. Differential analysis of the expression of 202 SFs using Wilcoxon rank sum test.** SF, splicing factor. (XLSX)

**S2 Data. Differential analysis of the methylation of 202 SFs using Wilcoxon rank sum test.** SF, splicing factor. (XLSX)

**S3 Data. List of significantly differentially included exons between tumor and normal samples from TCGA.** TCGA, The Cancer Genome Atlas.
(XLSX)

**S4 Data. List of 314 differentially included exons (ΔPSI ≥ 25) upon SRRM4 overexpression in HEK293.** PSI, percent spliced in; SRRM4, Serine/Arginine Repetitive Matrix 4.
(XLSX)

**S5 Data. GO term enrichment analysis of genes with SRRM4 target exons with decreased inclusion in tumors.** GO, gene ontology; SRRM4, Serine/Arginine Repetitive Matrix 4.
(XLSX)

**S6 Data. List of differentially included exons (ΔPSI ≥ 25) upon WT SRRM4 overexpression in MCF7, MDA-MB-231, DU145, HeLa, HCT116, and SH-SY5Y.** PSI, percent spliced in; SRRM4, Serine/Arginine Repetitive Matrix 4; WT, wild-type.
(XLSX)

**S7 Data. List of differentially expressed genes (fold change ≥2) upon WT SRRM4 overexpression in MCF7, MDA-MB-231, DU145, HeLa, HCT116, and SH-SY5Y.** Units of expression shown are cRPKM. SRRM4, Serine/Arginine Repetitive Matrix 4; WT, wild-type.
(XLSX)

**S8 Data. Binomial test results for associations of gene expression changes (|fold change| ≥2) between cell lines with SRRM4 overexpression (WT vs. DM) and a neuronal differentiation dataset (mature neurons vs. neuronal precursor cells).** DM, deletion mutant; SRRM4, Serine/Arginine Repetitive Matrix 4; WT, wild-type.
(XLSX)

**S9 Data. Binomial test results for associations of exon inclusion changes (|ΔPSI ≥ 5|) between cell lines with SRRM4 overexpression (WT vs. DM) and a neuronal differentiation dataset (mature neurons vs. neuronal precursor cells).** DM, deletion mutant; PSI, percent spliced in; SRRM4, Serine/Arginine Repetitive Matrix 4; WT, wild-type.
(XLSX)

**S10 Data. List of differentially expressed genes (|fold change| ≥2) upon WT SRRM4 induction in MDA-MB-231 xenografts in mice.** Units of expression shown are cRPKM. SRRM4, Serine/Arginine Repetitive Matrix 4; WT, wild-type.
(XLSX)

**S11 Data. Binomial test results for associations of exon inclusion changes (|ΔPSI| ≥ 5) between MDA-MB-231 xenografts after SRRM4 induction and a neuronal differentiation dataset (mature neurons vs. neuronal precursor cells).** PSI, percent spliced in; SRRM4, Serine/Arginine Repetitive Matrix 4.
(XLSX)

**S12 Data. Binomial test results for associations of gene expression changes (|fold change| ≥2) between MDA-MB-231 xenografts after SRRM4 induction and a neuronal differentiation dataset (mature neurons vs. neuronal precursor cells).** SRRM4, Serine/Arginine Repetitive Matrix 4.
(XLSX)

**S13 Data. Binomial test results for associations of gene expression changes (|fold change| ≥2) between cell lines with SRRM4 overexpression and an SH-SY5Y differentiation**

**dataset.** SRRM4, Serine/Arginine Repetitive Matrix 4.
(XLSX)

**S1 Data Figures. This file contains the data behind all figures in the main text and Supporting information as indicated in the figure legends.**
(XLSX)

**S1 Raw Images. This file contains the uncropped and unedited images of all gels shown in the manuscript figures indicated.**
(PDF)

## Acknowledgments

We thank Patryk Polińksi, Reza Sodaei, and Sophie Bonnal for advice and provision of reagents; Juan Valcarcel for critical reading of the manuscript; Joseph DiBartolo for assistance with data processing; and the CRG Genomics Unit for assistance with RNA sequencing services. The results published here are in part based upon data generated by the TCGA Research Network: https://www.cancer.gov/tcga.

## Materials and correspondence

Further information and requests for reagents may be directed to Luis Serrano (luis.serrano@crg.eu).

## Author Contributions

**Conceptualization:** Martin H. Schaefer, Luis Serrano.

**Data curation:** Jae-Seong Yang, Martin H. Schaefer.

**Formal analysis:** Sarah A. Head, Xavier Hernandez-Alias, Jae-Seong Yang, Antonio Torres-Méndez, Martin H. Schaefer.

**Funding acquisition:** Sarah A. Head, Luis Serrano.

**Investigation:** Sarah A. Head, Xavier Hernandez-Alias, Ludovica Ciampi, Violeta Beltran-Sastre, Antonio Torres-Méndez.

**Methodology:** Manuel Irimia.

**Project administration:** Luis Serrano.

**Software:** Manuel Irimia.

**Supervision:** Martin H. Schaefer, Luis Serrano.

**Validation:** Sarah A. Head, Xavier Hernandez-Alias, Martin H. Schaefer.

**Visualization:** Sarah A. Head, Xavier Hernandez-Alias.

**Writing – original draft:** Sarah A. Head, Martin H. Schaefer.

**Writing – review & editing:** Sarah A. Head, Xavier Hernandez-Alias, Martin H. Schaefer, Luis Serrano.

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
