## [Editor Report · Decision Letter 0]

14 Apr 2020

Dear Dr Serrano, 

Thank you for submitting your manuscript entitled "Hypersilencing of SRRM4 suppresses basal microexon inclusion and promotes tumor growth across cancers" for consideration as a Research Article by PLOS Biology.

Your manuscript has now been evaluated by the PLOS Biology editorial staff as well as by an academic editor with relevant expertise and I am writing to let you know that we would like to send your submission out for external peer review.

Please re-submit your manuscript within two working days, i.e. by Apr 16 2020 11:59PM.

Kind regards,

Hashi Wijayatilake, PhD,

Managing Editor

PLOS Biology

---

## [Decision Letter · Decision Letter 1]

8 Jun 2020

Dear Dr Serrano,

Thank you very much for submitting your manuscript "Hypersilencing of SRRM4 suppresses basal microexon inclusion and promotes tumor growth across cancers" for consideration as a Research Article at PLOS Biology. Your manuscript has been evaluated by the PLOS Biology editors, an Academic Editor with relevant expertise, and several independent reviewers. We thank you again for your patience and understanding during the delays.

As you will see from the comments below, the reviewers find the manuscript interesting and intriguing. Reviewer 1 and 2 do however have a number of importance concerns that should be addressed. In particular, Rev. 2 questions the functional relevance of the transcript-level effects observed and Rev. 1 questions why altering SRRM4 levels and microexon inclusion would alter tumor cell growth. We will not be able to accept the current version of the manuscript, but we would welcome re-submission of a much-revised version that takes into account the reviewers' comments. The Academic Editor has provided helpful guidance regarding the requests and revisions, which can be found below the reviews.

Please note that we cannot make any decision about publication until we have seen the revised manuscript and your response to the reviewers' comments. Your revised manuscript is also likely to be sent for further evaluation by the reviewers.

We expect to receive your revised manuscript within 2 months. Please email us (plosbiology@plos.org) if you have any questions or concerns, or would like to request an extension. At this stage, your manuscript remains formally under active consideration at our journal; please notify us by email if you do not intend to submit a revision so that we may end consideration of the manuscript at PLOS Biology.

**IMPORTANT - SUBMITTING YOUR REVISION**

*Re-submission Checklist*

*Published Peer Review*

*PLOS Data Policy*

*Blot and Gel Data Policy*

Sincerely,

Hashi Wijayatilake, PhD, 

Managing Editor

PLOS Biology

REVIEWS:

Reviewer #1: 

This is an interesting manuscript which identifies that the splicing factor SRRM4 is greatly silenced in tumor cells compared to normal cells. SRRM4 is a regulator of microexon inclusion and inclusion of SRRM4 target exons is generally downregulated in tumors. These results are novel. The major weakness of the manuscript are as follows:

1) It is not clear why altering SRRM4 levels and microexon inclusion would alter tumor cell growth. Are there specific microexons whose inclusion are critical to the growth regulatory effects of altering SRRM4 levels? 

2) It would be helpful to show a few examples of splicing events (eg by RNA-seq coverage plots, Sashimi plots, or RT-PCR for example) regulated by SRRM4 in the cell line experiments performed here. 

3) Can the authors demonstrate that methylation of the SRRM4 promoter reduces SRRM4 mRNA/protein levels? 

4) Does downregulation of SRRM4 in non-malignant, non-neuronal cells alter their growth?

5) The suggestion that altering SRRM4 levels in somatic cancer cells makes them "neuron-like" seems to be a stretch for non-neuronal cells. The idea that a transformed epithelial will grow projections that having anything to do with axons by altering SRRM4 levels is hard to believe. This point needs to be softened in the manuscript. 

---

Reviewer #2: 

Head et al. investigate the role of SRRM4, a neuronal splicing factor that regulates micro-exons, in human cancers. Although SRRM4 has been implicated in neuronal tissue development and in neuronal diseases, whether SRRM4 can contribute to disease pathogenesis in non-neuronal tissue was unknown. Here, the authors analyze RNA-seq data from >6,900 TCGA cancer patients and demonstrate that SRRM4 is expressed at lower levels and is hypermethylated in tumor vs. adjacent normal tissues. They identify a small set of SRRM4-regulated micro-exons that are differentially spliced in several tumor types. They show that overexpression of SRRM4 WT but not of a deletion mutant decreases growth of cancer cell lines in vitro, as well as tumor xenograft growth in vivo in a breast cancer murine model. Finally, they uncover a set of exons more included across multiple cancer cell lines with SRRM4 overexpression. 

Overall, the findings presented here are interesting, represent a considerable amount of work, and provide the first evidence of the molecular role of SRRM4 mutants in non-neuronal tissues. However, the experimental design, computational and statistical analysis are insufficiently described, making it difficult to fully grasp the validity of some of the study findings. In addition, several key questions should be addressed before the proposed research is ready for publication. Importantly, the paper is based on the observation that SRRM4 transcripts are expressed at lower levels in tumors vs. normal tissue, that relies on data that most scientists would consider as noise in RNA-seq experiments. Therefore it is critical that the authors demonstrate that this level of change in transcript expression leads to protein changes in tumors and has a functional consequence in cell lines. Their experiments use overexpression at >20-100 fold protein changes that are likely not to be physiologically relevant given the baseline SRRM4 expression in tumors and normal tissues.

Major comments:

1) As stated by the authors and shown in Fig. 2, SRRM4 transcripts are expressed at very low levels in non-neuronal tissues (median RSEM expression ~4 in normal tissues, ~0.6 in tumors). Those values are typically filtered out from a differential expression analysis because of the low confidence of these reads which are usually considered as noise. As a comparison median expression of SSRM1 transcripts is ~1,500, SSRM2 ~10,000, SRRM3 ~50 RSEM counts This raises the question of the sensitivity and accuracy of SRRM4 transcripts detection using RNA-seq data, especially considering that TCGA data has overall rather poor read coverage. Therefore, one wonders if these small changes in SRRM4 transcript levels can actually have any physiological consequences and lead to detectable changes in SRRM4 protein expression in tumors vs. normal tissues. This is a critical question as all the paper relies on this assumption.

The authors compare expression in tumors vs. normal tissues, yet the details of how the differential analysis was performed or what is the magnitude of fold change are not provided; these details should be provided. Fig 2A should also indicate whether the data is plotted as median or mean, and what the quartile ranges are. 

Additionally, the authors claim that other SRRM genes (SRRM1-3) are not differentially expressed in tumors. Yet the data presented in Dataset 1 show significant P-values, even lower than for SRRM4, for SRRM1-3 transcripts in many other tumors types. The authors need to clarify this discrepancy between the data presented in the Fig 2C and Dataset 1. 

Given the low baseline SRRM4 transcripts expression, even a slight change in counts would be accounted as a major expression change. Therefore, for Fig. 2D, one wonders if SRRM4 is the top hit because if its low expression. We recommend the authors demonstrate that other splicing-factor genes expressed at low levels (within the same RSEM values) in non-neuronal tissues do not exhibit expression differences between tumors vs. normal tissues.

2) Although SRRM4 is mostly expressed in neuronal cell types, the author focus their analysis on non-neuronal tumor types. Yet they state "the most striking decrease was observed in glioblastoma, likely due to the higher inclusion of SRRM4 targets in glia than other normal cell types", suggesting that SRRM4 might play a role in tumors of neuronal origin. Given the availability of RNA-seq data from neuronal tumors in TCGA, as well as normal samples in GTEX, the authors should explain why they did not investigate the role of SRRM4 in these tumors. The author state they use RNA-seq data from TCGA for which paired normal samples are available and therefore would exclude neuronal TCGA tumors. Yet since they do not performed paired analysis for each patient, they could very well include data from normal GTEX neuronal tissues, which would permit the analysis of neuronal tumor types for which no TCGA data from normal tissue would be available. 

3) The authors focus on a set of spliced isoforms regulated by SRRM4 in neuronal cell types (Fig. 3). Whether these exons are relevant in non-neuronal cells remains to be determined. First, the authors should explain where does the list of SRRM4 targets comes from? The authors claim that tumor spliced exons are "significantly enriched" in SRRM4 targets. How many of these are micro-exons? Fig. 3A seems to indicate <50 SRRM4 targets are spliced-out in tumor vs. normal, which is not very impressive, especially given that none of them is detected across all tumor types. 

The authors perform differential splicing analysis on tumor vs. normal samples using a cut-off of >5% splicing change is within the limit of biological and technical noise. Yet, when analyzing data from cell lines they use a cut-off of >25%. We recommend using the same cut-off measuring at least >10% of splicing change differences.

Given the known genomic intra- and inter-tumor heterogeneity, and since Fig. 2A indicates widespread differences in SRRM4 transcripts levels within tumors, the authors could have divided the tumors into low- and high-SRRM4 expression, and compared the splicing targets within these two classes. This approach could potentially additional SRRM4 targets that are not diluted in a mixture of low- and high-SRRM4 expressing samples. 

Also, the authors should consider the possibility that these spliced isoforms can be regulated by other splicing factors, for example SRRM3 which regulate very similar targets to SRRM4 (Gonatopoulos-Pournatzis et al. 2018), as well as SRSF11, RBFOX or PTBP1 which also regulate micro-exons (Gonatopoulos-Pournatzis et al. 2018, Li et al, 2015). Whether these genes are differentially expressed along with SRRM4 in tumors vs. normal tissues should be determined. Finally, given the fact that 2-fold changes in SF-levels are sufficient to lead to splicing changes, the authors should at least discuss the possibility that other SFs, including SRNPB, RBM34, RNPS1, ELAVL1 an RALYL, could contribute to tumor-associated splicing events in the analyzed datasets.

4) Does SRRM4 regulate other exons that are not micro-exons? The author imply this but don't say it. Isn't it possible that these non-microexons are contributing to phenotype? The splicing analysis in tumors should highlight separately the findings from micro-exons and non micro-exons in Fig 3. Similarly, one wonders if the gene enrichment analysis in Fig 4 relates to all SRRM4 targets or just microexons?

5) In Fig 4, the authors correlate SRRM4 transcript expression with the mitotic index, but fail to indicate whether SRRM4 expression correlates with other tumor features, including tumor grade, presence of know cancer driver alterations (e.g., TP53, PTEN, MYC etc), patient overall survival, as well as tumor purity. This last point is particular relevant given high intra-tumor heterogeneity in solid cancers, raising the question whether decreases SRRM4 expression detected in tumor samples is due to changes in stromal or immune cell composition in tumors vs. normal tissues rather than in cancer epithelial cells. 

6) In Fig. 5-7, the authors test the consequences of SRRM4 overexpression in cancer cell lines. Do these cell lines all have virtually no SRRM4 protein expression at baseline? It would be nice to see qPCR so that we can compare with the fold changes in tumors vs. normal tissues to the levels of overexpression achieve in the cell lines. If there is no baseline expression at all, then the levels of overexpression are way beyond physiological, and way beyond the changes detected in tumors. Since splicing factors act in dose dependent manner, this raises the question whether this level of expression is physiologically relevant and how it really mimics the changes between tumor and normal tissues. Indeed, in all cell lines, SRRM4 induction with 62 or 125 ng/ml of DOX leads to very mild phenotypic differences. Thus relevance of this data is difficult to grasp in the absence of a quantification of the SRRM4 protein expression in tumors vs. normal tissues. Furthermore, a western at baseline and one of the DOX concentration for all cell lines on one blot is needed so one can compare the expression levels and the effects on cell growth across multiple cell lines as presented in Fig 5E. The levels look widely different in Fig S5, raising the question whether difference in Fig 5E are due to differences in SRRM4 levels. Finally, the level of expression of SRRM4 WT and SRRM4 DM are different in Fig 5D, raising the question whether the lack of effect with SRRM4 in vitro and in vivo is due to lower expression. 

Finally, the authors should provide RT-PCR or QPCR validations for their top SRRM4-regulated splicing events in cancer cell lines to demonstrate the accuracy of their splicing analysis, and illustrate the false positive detection rate.

7) Comparing expression and splicing changes DOX treated vs. DOX untreated tumor samples can be confounded by DOX-induced expression and splicing as previously described (Ahler et al. 2013, Park et al. 2019). Therefore, events induced by DOX in empty vector control should be removed from the dataset in Fig 7f.

8) The author claim that SH-SY5Y cells expressing SRRM4 form "dendrite-like" projections, "reminiscent of SH-SY5Y cells that have undergone neuronal differentiation with retinoic acid (RA) and brain-derived neurotrophic factor (BDNF) treatment" based on a representative bright field picture. Have the authors used any neuronal differentiation cell markers to demonstrate that these are "dendrite-like"?

Minor comments

1) Figures 1-7 legends should be updated to include a better description of the data presented (mean, medina, range, etc.) as well as statistical test and relevant p-values and symbols shown in the figures. Figure labels and graph axes should be made readable throughout the text, and axes should be labelled with better descriptions. For example Fig. 5E what exacly is "Max % inhibition (4days)?"

2) Fig. 2E shows increased fold change in SSRM4 expression in tumors vs. normal tissues, yet the remaining figures show decrease expression. The scale/legend should be probably be corrected.

3) Fig. 2 - where does the list of 202 splicing factors comes from?

4) Fig. 3A the dual scale is confusing; this should be plotted separately. What is the dark red color?

5) Fig. 3D - please clarify what this figure is comparing cancer cell lines vs. non transformed cell lines? If this is cancer cell lines vs. normal tissues, then this comparison is not acceptable as cell lines are known to be different from primary tissues and exhibit distinct gene expression and splicing profiles. 

6) The authors state "The resulting list comprised between 468 exons in head and neck cancer to 1,884 in lung cancer, of which between 6.6% (head and neck cancer) and 12.7% (colorectal cancer) were micro-exons". Is the 12.7% in colorectal or lung? Is this a typo?

7) The level of SRRM4 in vivo in tumor xenograft experiment should be shown in Fig 6.

8) Fig. 7C needs statistics and the results of the differential gene expression analysis should be provided as table.

9) Supplemental Fig. 6 is cited but it is not in the document.

10) The discussion states "Notably, we find that prostate cancers and lung cancers taken as a whole follow the opposite trend as their neuroendocrine subtypes." - this is a potentially very exciting statement, but one would love to see that data in the paper.

---

Reviewer #3 (Steve Mount): 

This paper presents evidence that SRRM4 silencing in tumors promotes growth and is associated with reduced inclusion of microexons. The two principal claims (first, that SRRM4 is silenced in tumors; second, that this decreased expression affects growth and induction of SRRM4 expression inhibits proliferation) are well supported.

The paper is well-written. I read it very carefully and have no significant suggestions for revision (which is extremely unusual for me).

The one suggestion I would make (not mandatory) is to consider logit(PSI) in order to identify those events whose altered splicing occurs in the context of nearly complete skipping or nearly complete inclusion (delta-logit(PSI) can be significant when delta-PSI is not).

---

ACADEMIC EDITOR COMMENTS:

1) Referee 2, point 1: This is a key point. The SRRM4 levels indeed are very low and the authors need to provide convincing evidence that the observed differences between normal and tumor are not noise. The referee suggests several ways to do this: 1) compare SRRM4 changes to changes of other very low abundance transcripts, 2) demonstrate changes at the protein level and 3) provide more detailed statistical analysis to demonstrated significance. Showing only 3) would not be sufficient. 

2) Referee 2, point 4: This is another key point. There is the possibility that the observed phenotypes are due to effects that occur via SRRM4 but not via micro-exons. The authors should demonstrate and discuss the effect of SRRM4 on exons other than micro-exons and provide evidence that they do not contribute to phenotype, for example, by demonstrating that non-microexon events do not occur in cancers.

3) Physiological relevance: This is a difficult point to fully resolve, but I agree with Referee 1, point 4 that it would be reassuring to see that downregulation in non-malignant cells has an effect. The flipside of this is referee 2, point 6, highlighting that the overexpression experiments compare apples to oranges by introducing SRRM4 at much higher levels than physiologically observed, making its interpretation difficult. The suggested PCR analysis would give a better sense of how far off the comparison is.

---

## [Decision Letter · Decision Letter 2]

7 Jan 2021

Dear Luis,

Thank you for submitting your revised Research Article entitled "Hypersilencing of SRRM4 suppresses basal microexon inclusion and promotes tumor growth across cancers" for publication in PLOS Biology. Thank you also for your patience as we completed our editorial process, and please accept again my apologies for the delay in providing you with our decision. I have now obtained advice from two of the original reviewers and have discussed their comments with the Academic Editor. 

Based on the reviews, we will probably accept this manuscript for publication, assuming that you will modify the manuscript to address the remaining points raised by Reviewer 2. Please also make sure to address the data and other policy-related requests noted at the end of this email.

We expect to receive your revised manuscript within two weeks. Your revisions should address the specific points made by each reviewer. 

-  a cover letter that should detail your responses to any editorial requests, if applicable

*Published Peer Review History*

*Early Version*

Best wishes,

Ines

--

Ines Alvarez-Garcia, PhD,

Senior Editor,

ialvarez-garcia@plos.org,

PLOS Biology

ETHICS STATEMENT:

-- Thank you for including the name of the ethics committe. Please also include the license or approval number.

DATA POLICY:

Many thanks for submitting the data file to comply with our data policy, however we are still missing the data underlying the following figures:

Fig. 2C-G; Fig. 3A, B, D; Fig. 4A-D; Fig. 5A-C, E; Fig. 7D, E; Fig. S2; Fig. S3; Fig. S5; Fig. S7; Fig. S9 and Fig. S10A, B

Please provide it and indicate in the figure legends where the underlying data can be found, and ensure your supplemental data file/s has a legend. If any of the data are already deposited in a database, please indicate this clearly also in the figure legends.

Reviewers’ comments

Rev. 1:

The authors have addressed my prior concerns and the manuscript is improved. I have no further issues with the manuscript.

Rev. 2:

The authors have considerably revised and improved the manuscript.

A few points should still be clarified before publication.

Major comments

1) The author added a new experiment in Fig. 5e-f. The authors state they mixed WT or DM SRRM4 cells at a 1:1 ratio, yet at day 1 the ratio detected by PCR is not 1:1, raising the question of how quantitative and reproducible this assay is. Moreover, this experiment shows only one replicate and no control for the effects of doxycycline. Finally, the interpretation of this growth competition assay in Fig.5, as well as the xenograft assays in Fig.6, is challenging because the DM cell line grows more slowly, even without DOX (induced), compared to the WT cell line. As show in Fig. 6, uninduced WT and DM cells reach a tumor volume of a 100 at 20 and 40 days respectively. Therefore experiments in Fig. 5e-f should either be removed, or repeated with appropriate controls. The current version shows the growth difference between the two clones, and does not point out necessarily to an effect of SRRM4.

2) In Fig. 7 from their response, the authors show that indeed the level of expression of SRRM4 WT and SRRM4 DM are different at the same doxycycline concentration. Therefore they should acknowledge in the text that the lack of effect of the DM SRRM4 on cell proliferation could be due to lower expression.

Minor comments

1) As part of best reproducibility practices, please add versions of computation tools, links to github pages if available, and specific parameters used.

2) The word 'hypersilenced' should be replaced by 'silenced' throughout the text and title.

3) The title should be edited to reflect more accurately the findings described in the paper. What is the meaning of basal micro-exons in the title?

---

## [Editor Report · Decision Letter 3]

3 Feb 2021

Dear Luis,

Thank you for submitting your new revised Research Article entitled "Silencing of SRRM4 suppresses microexon inclusion and promotes tumor growth across cancers" for publication in PLOS Biology. I have now discussed it with the Academic Editor. 

While we are mostly satisfied with your responses to the pending issues, we would like you to include in the Figure 5 legend a statement indicating the reason for the deviation from the expected 1:1 ratio in the PCR reaction. We think it is a fairly obvious discrepancy and readers may wonder about it.

We expect to receive your revised manuscript within one week.

Best wishes,

Ines

--

Ines Alvarez-Garcia, PhD,

Senior Editor,

PLOS Biology

---

## [Editor Report · Decision Letter 4]

4 Feb 2021

Dear Luis,

On behalf of my colleagues and the Academic Editor, Tom Misteli, I am pleased to say that we can in principle offer to publish your Research Article entitled "Silencing of SRRM4 suppresses microexon inclusion and promotes tumor growth across cancers" in PLOS Biology, provided you address any remaining formatting and reporting issues. These will be detailed in an email that will follow this letter and that you will usually receive within 2-3 business days, during which time no action is required from you. Please note that we will not be able to formally accept your manuscript and schedule it for publication until you have made the required changes.

PRESS

Thank you again for supporting Open Access publishing. We look forward to publishing your paper in PLOS Biology. 

Sincerely, 

Ines

--

Ines Alvarez-Garcia, PhD 

Senior Editor 

PLOS Biology
